# Learning in the Fisher Subspace: A Guided Initialization for LoRA Fine-Tuning

**Zhi-Quan Feng** [1]   **Ying-Jia Lin** [2]   **Hung-Yu Kao** [3]

## Abstract

LoRA adapts large language models (LLMs) by restricting updates to low-rank subspaces of pre-trained weights. While this substantially reduces training cost, the effectiveness of adaptation critically depends on which subspace is chosen at initialization: a poor initialization that allocates capacity to task-irrelevant directions can severely hinder downstream performance. Existing initialization strategies primarily rely on the intrinsic properties of pre-trained weights, implicitly assuming that weight geometry alone reflects task relevance. However, such criteria overlook how the model interacts with the downstream data distribution. In this work, we formulate LoRA initialization as identifying the degree of impact of directions in parameter space under the target data distribution. We argue that data-aware sensitivity, rather than weight-only magnitude, should govern the choice of adaptation subspaces. Building on this perspective, we propose a Fisher-guided framework that leverages curvature information induced by downstream data to characterize how parameter perturbations influence model predictions. This perspective yields a principled, task-dependent criterion for selecting LoRA directions that better align adaptation with the target objective. Empirical results across diverse tasks and modalities demonstrate that data-aware initialization consistently and significantly improves downstream performance over existing approaches.

## 1. Introduction

Parameter-Efficient Fine-Tuning (PEFT) methods, especially LoRA, offer a practical alternative by injecting small, trainable low-rank adapters into frozen weights of large language models (LLMs) to reduce the computation cost of training LLMs. Beyond reducing parameter counts, the effectiveness of such adapters depends critically on which subspace of the original weight space they occupy: a well-chosen initialization can concentrate learning on task-relevant degrees of freedom and drastically improve downstream task convergence as well as final accuracy.

Most advanced LoRA initialization strategies are grounded in the geometric structure of pre-trained weights, often operationalized through singular value decomposition (SVD) of the corresponding weight matrices (Meng et al., 2024; Wang et al., 2025b;a). While spectral decompositions of pre-trained weights reveal important structural priors, they are blind to the downstream data distribution and the sensitivity of the model to that data. Recent empirical studies show that the singular directions most useful for one task can be uninformative for another (Fan et al., 2025). In short, weight-only criteria risk allocating scarce adaptation capacity to directions that are passive under the target data distribution.

A more principled approach is to select adaptation directions using data-aware statistics that quantify how parameter perturbations influence the model's outputs on the target task. We define Fisher energy based on second-order derivatives of the weight matrices, and our observation shows that Fisher Energy provides an effective measure of parameter sensitivity with respect to the data distribution, and that prioritizing directions with lower Fisher Energy leads to improved downstream performance.

Motivated by this observation, we introduce FILet (Fisher-Guided Initialization for LoRA Fine-Tuning), a data-aware framework that leverages Fisher Energy to identify and align LoRA adaptations with convergence-favorable directions in parameter space. Specifically, FILet computes the second order derivation using Kronecker-factored empirical covariances, which enables efficient capture of data-induced parameter sensitivity without incurring significant computational overhead. Fisher Energy is then used to guide the initialization of LoRA factors, constraining parameter updates to Fisher-aligned subspaces. We comprehensively evaluate FILet against existing SVD-based (Meng et al., 2024; Wang et al., 2025b;a) and data-driven LoRA initialization

---

[1] Department of Computer Science and Information Engineering, National Cheng Kung University [2] Department of Artificial Intelligence, Chang Gung University [3] Department of Computer Science, National Tsing Hua University. Correspondence to: Hung-Yu Kao <hykao@cs.nthu.edu.tw>.

*Proceedings of the 43rd International Conference on Machine Learning*, Seoul, South Korea. PMLR 306, 2026. Copyright 2026 by the author(s).

methods (Si et al., 2025) across natural language reasoning, image classification, and natural language generation tasks on large language models. Experimental results demonstrate that this data-aware initialization strategy consistently and significantly improves overall performance across diverse downstream tasks.

## 2. Related Work

LoRA (Hu et al., 2022) has emerged as one of the most popular and powerful PEFT techniques for LLMs. By introducing trainable low-rank matrices to approximate weight updates, LoRA substantially reduces the number of trainable parameters while maintaining performance comparable to full fine-tuning. Building upon this foundation, numerous LoRA variants have been proposed to further enhance adaptability and efficiency from various perspectives, including dynamic rank adaptation (Zhang et al., 2023), rank expansion through nonlinear transformations (Ji et al., 2025), reduction of trainable parameters (Kopiczko et al., 2024; Zhang et al., 2025b), multitask learning extensions (Yang et al., 2025; Zhang et al., 2025a), and integration with mixture-of-experts (MoE) architectures (Tian et al., 2024; Zadouri et al., 2024; Fan et al., 2025).

One of the most actively explored research directions in LoRA-based fine-tuning concerns the adaptation direction of low-rank modules. DoRA (Liu et al., 2024) first highlighted this issue by decomposing LoRA updates into magnitude and direction components, demonstrating that tuning only the directional component can yield more stable and efficient adaptation. Other methods such as PiSSA (Meng et al., 2024), MiLoRA (Wang et al., 2025b), CorDA (Yang et al., 2024), and KaSA (Wang et al., 2025a) further investigate direction-aware adaptation by decomposing pre-trained weights via SVD. These approaches define and select the most important singular directions to initialize the LoRA parameters, while retaining other directions in the frozen residual weights. Suppose a weight matrix $W_0$ has the SVD decomposition $W_0 = U\Sigma V^\top$, where $U$ and $V$ are orthonormal matrices and $\Sigma$ is a diagonal matrix of singular values. The LoRA Modules $A \in \mathbb{R}^{r \times n}$ and $B \in \mathbb{R}^{m \times r}$ are then initialized as:

$$A = \sqrt{\Sigma'}\, V'^\top,\ B = U'\sqrt{\Sigma'},\ W_{\text{res}} = W_0 - AB, \quad (1)$$

where $U'$, $V'$, and $\Sigma'$ correspond to the selected singular directions and values based on the specific criteria of each method. $W_{\text{res}}$ denotes the residual weights and $W_0$ denotes the pre-trained weights.

PiSSA focuses on fine-tuning the most principal singular directions, capturing the most informative subspace of the pre-trained weights. In contrast, MiLoRA targets the least dominant singular directions, emphasizing the underrepresented components ignored during pre-training. KaSA

further refines this idea by discarding the least dominant directions to reduce noise, and selectively training on moderately weak directions that preserve informative variability for LoRA adaptation.

However, a key limitation of these approaches lies in their exclusive focus on interpreting the spectral properties of pre-trained weights, which may not faithfully capture the model's sensitivity to data. This raises a natural question: *Why not directly ask the data?* In other words, downstream data-aware method is more principled than these heuristic weight-only criteria for selecting adaptation directions.

Current data-aware methods, such as LoRA-Dash (Si et al., 2025), proposes a training-time enhancement for LoRA: it performs a short warm-up fine-tuning, then applies SVD to the LoRA weights to extract task-specific directions (TSD) for reparameterization, aiming to accelerate convergence and improve performance. LoRA-GA (Wang et al., 2024) and EVA (Paischer et al., 2025) also leverage data-driven statistics to guide LoRA adaptation. LoRA-GA aligns low-rank updates with full fine-tuning gradients via direction selection and scaling while EVA performs SVD on minibatch activation vectors and emphasizes explained variance and rank reallocation.

## 3. Method

In this section, we introduce the **Fisher-Guided LoRA Initialization** framework. Our goal is to construct low-rank adaptation matrices whose initialization directions are primarily determined by data sensitivity rather than singular values. We present the problem definition in *3.1 Problem Definition* and our preliminary experiments for this design in *3.2 Preliminary*.

Our method, illustrated in Figure 2, is introduced in Section 3.3 *Fisher-Guided Initialization*. Appendix A further summarizes the symbols and notations used throughout the paper, presents the algorithm pseudocode, and provides additional mathematical derivations and proofs.

### 3.1. Problem Definition

The LoRA initialization problem can be formulated as partitioning the pretrained weight space into two complementary subspaces: one corresponding to the frozen residual weights, and the other to the trainable low-rank adapters. Our objective is to identify directions that carry minimal information relevant to the downstream task, and to initialize the LoRA modules along these directions, thereby preserving task-relevant knowledge within the residual weights. Intuitively, directions that induce a large increase in loss are likely to encode important task-specific information and should therefore be preserved. Accordingly, we aim to minimize the risk of loss increase induced by perturbations from the LoRA

modules.

Our complete set of symbols and notations is summarized in Appendix A.1. Let $W \in \mathbb{R}^{m \times n}$ denote the weight matrix to be adapted for a downstream task. In the LoRA framework, the adaptation is parameterized by two low-rank matrices $A \in \mathbb{R}^{r \times n}$ and $B \in \mathbb{R}^{m \times r}$, where $r \ll \min(m, n)$. Our data-driven initialization strategy aims to identify task-relevant subspaces of $W$ and leverage them to guide the low-rank adaptation.

Specifically, we focus on selecting a group of rank-1 directions, each direction $Z$ is defined as:

$$Z := \left\{ uv^\top \,\middle|\, u \in \mathbb{R}^m, v \in \mathbb{R}^n, \|u\|_2 = 1, \|v\|_2 = 1 \right\}, \quad (2)$$

where $u$ and $v$ are used to initialize $B$ and $A$, respectively. For a small scalar $\gamma > 0$, perturbing the pre-trained weights along such a direction yields the updated weights $W_{\text{updated}} = W_0 + \gamma Z$. We quantify the utility of a direction $(u, v)$ by the corresponding change in the downstream loss,

$$\Delta \mathcal{L}(Z) := \mathcal{L}(W_0 + \gamma Z) - \mathcal{L}(W_0), \quad (3)$$

where $\mathcal{L}$ denotes the task-specific loss function. Our objective is to select directions $(u, v)$ that minimize the expected increase in loss, i.e., those that most effectively reduce $\Delta \mathcal{L}$ in expectation. By applying a second-order Taylor expansion of the loss function around $W_0$, $\mathcal{L}(W_0 + \gamma Z) - \mathcal{L}(W_0)$ can be expressed as:

$$\begin{aligned} \mathcal{L}(W_0 + \gamma Z) - \mathcal{L}(W_0) = \; & \gamma \left\langle \nabla_{W_0} \mathcal{L}, Z \right\rangle \\ & + \frac{\gamma^2}{2} \left\langle Z, \left( \nabla_{W_0}^2 \mathcal{L} \right) Z \right\rangle \\ & + o(\gamma^2), \end{aligned} \quad (4)$$

where $\nabla_{W_0} \mathcal{L}$ and $\nabla_{W_0}^2 \mathcal{L}$ denote the gradient and Hessian of the loss with respect to $W_0$, respectively, and $\langle \cdot, \cdot \rangle$ represents the Frobenius inner product. The equation 4 decomposes the loss change into two components: The first-order term captures the gradient direction, while the second-order term characterizes the curvature, or sensitivity, of the loss landscape along direction $Z$. Our objective is to identify directions that contain minimal information relevant to the downstream task and utilize the signal that cannot be captured during fine-tuning. Under this perspective, the second-order term becomes the primary factor, as a direction with high sensitivity which cause a larger increase in loss is imply that it contains more task-relevant information. Therefore, we adopt symmetric perturbation to isolate the second-order term, which can be expressed as:

$$\begin{aligned} \Delta \mathcal{L}_{\text{sym}}(Z) := \; & \frac{1}{2} \left[ \mathcal{L}(W_0 + \gamma Z) + \mathcal{L}(W_0 - \gamma Z) \right] - \mathcal{L}(W_0) \\ = \; & \frac{\gamma^2}{2} \left\langle Z, \left( \nabla_{W_0}^2 \mathcal{L} \right) Z \right\rangle + o(\gamma^2). \end{aligned} \quad (5)$$

Under this symmetric construction, the first-order terms cancel out by design, yielding:

$$\Delta \mathcal{L}_{\text{sym}}(Z) \approx \frac{\gamma^2}{2} \left\langle Z, \left( \nabla_{W_0}^2 \mathcal{L} \right) Z \right\rangle. \quad (6)$$

This formulation focuses on the curvature of the loss landscape. By selecting $Z$ along the top singular directions of $W_0$, we bias the LoRA updates toward directions that are empirically most effective for facilitating loss convergence in the pre-trained model. According to equation 6, and inspired by prior work (Pascanu & Bengio, 2014), we approximate the Hessian in the second-order term along a given direction using the Fisher information matrix, defined as

$$S_W \; = \; \mathbb{E} \left[ \text{vec} \left( \nabla_W \mathcal{L} \right) \text{vec} \left( \nabla_W \mathcal{L} \right)^\top \right]. \quad (7)$$

Again following equation 6, the expected increase in loss caused by a small perturbation along direction $Z$ can be approximated by

$$\Delta \mathcal{L}_{\text{sym}}(Z) \; \propto \; \frac{\gamma^2}{2} \text{vec}(Z)^\top S_W \text{vec}(Z). \quad (8)$$

This observation motivates defining the Fisher Energy of a direction $Z$ as

$$\begin{aligned} \mathcal{E}(Z) := & \text{vec}(Z)^\top S_W \text{vec}(Z) \\ = & \text{vec}(uv^\top)^\top S_W \text{vec}(uv^\top). \end{aligned} \quad (9)$$

The Fisher Energy quantifies the curvature of the loss landscape along the direction $Z$, and equivalently measures the local sensitivity of the model's predictive distribution to perturbations in this direction. To minimize the expected loss increase $\Delta \mathcal{L}$, we therefore seek directions with low Fisher Energy $\mathcal{E}(Z)$ for LoRA initialization.

### 3.2. Preliminary

To further evaluate the effectiveness of Fisher Energy, we conduct preliminary experiments by selecting 32 distinct singular directions associated with different singular values. Notably, we separately investigate the effects of direction selection and direction scaling. Specifically, when evaluating the effect of direction selection, we control for direction scaling by uniformly using either the maximal or the minimal singular values across all selected directions. Accordingly, for the $i^{\text{th}}$ group, we initialize the LoRA modules as

$$\begin{aligned} A &= \sqrt{\Sigma_{(\text{min/max})}} \; V_{\frac{ih}{32} : \frac{ih}{32} + r}, \\ B &= U_{\frac{ih}{32} : \frac{ih}{32} + r} \; \sqrt{\Sigma_{(\text{min/max})}}, \\ W_{\text{res}} &= W_0 - AB, \end{aligned} \quad (10)$$

where $U, \Sigma, V$ are the SVD results and $h$ denotes the total number of singular directions, and $\Sigma_{(\text{min/max})}$ indicates that

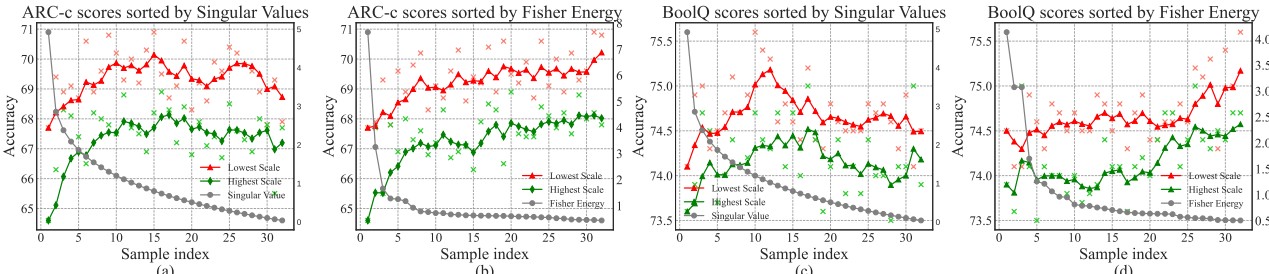

*Figure 1.* Experiments comparing singular-direction selection and magnitude-scaling strategies for LoRA initialization. For the 32 samples, panels (a) and (c) sort directions by singular values, while (b) and (d) sort them by their Fisher Energy values. Results are obtained on ARC-Challenge and BoolQ using Llama2-7B with rank = 32. The horizontal axis denotes the index of the sorted experiments. Scatter points show the raw experimental results, while the red and green curves indicate trends smoothed by exponential moving average (EMA).

either the minimal or the maximal singular values are used to scale each selected direction.

The experimental results are presented in Figure 1, where the outcomes are sorted by singular values and Fisher Energy, respectively. The results show that samples achieving higher downstream performance do not necessarily rely on directions associated with smaller singular values. Instead, these samples tend to correspond to directions with lower data sensitivity. In other words, data sensitivity is an important factor driving the performance differences and it provides a more informative ranking signal than singular values alone. Moreover, direction scaling plays a crucial role in downstream performance. we observe that minimal direction scaling consistently outperforms maximal direction scaling, irrespective of the direction-selection strategy. As a result, we conclude that we should prioritize low-Fisher-Energy directions and apply minimal scaling when initializing LoRA modules.

However, computing the full Fisher information matrix is computationally prohibitive for large-scale models due to its massive size in practice, as shown in Appendix A.3. Consequently, we propose a practical and efficient method to approximate Fisher information and select low-Fisher-energy directions for LoRA initialization, thereby enhancing fine-tuning performance across diverse downstream tasks.

### 3.3. FILet

For a linear transformation $Y = WX$, the Fisher information of the weight matrix $W \in \mathbb{R}^{m \times n}$ is defined as

$$S_W = \mathbb{E}\left[\operatorname{vec}\left(\nabla_W \mathcal{L}\right) \operatorname{vec}\left(\nabla_W \mathcal{L}\right)^\top\right],$$
$$\nabla_W \mathcal{L} = (\nabla_Y \mathcal{L})X^\top. \quad (11)$$

We leverage the K-FAC approximation (Martens & Grosse, 2015), which factorizes the gradient as $\nabla_W \mathcal{L} = X \otimes \nabla_Y \mathcal{L}$ (as proved in Appendix A.4), to express the Fisher informa-

tion as

$$S_W = \mathbb{E}\left[\left(X \otimes \nabla_Y \mathcal{L}\right)\left(X \otimes \nabla_Y \mathcal{L}\right)^\top\right]. \quad (12)$$

Under the assumption that $X$ (the input samples) and $\nabla_Y \mathcal{L}$ (the output gradients, which depend on the model architecture) are independent in LLMs, the expectation factorizes into a Kronecker product of second-moment matrices:

$$S_W \approx S_X \otimes S_Y,$$
$$S_X = \mathbb{E}\left[XX^\top\right],$$
$$S_Y = \mathbb{E}\left[(\nabla_Y \mathcal{L})(\nabla_Y \mathcal{L})^\top\right]. \quad (13)$$

This Kronecker-factorized form enables efficient computation of Fisher geometry using minibatch estimates, without forming the full $(mn) \times (mn)$ Fisher matrix. To compute $S_X$ and $S_Y$, we sample a minibatch of data from the downstream data distribution and collect the corresponding input activations $X$ and output gradients $\nabla_Y \mathcal{L}$.

We next select the singular bases according to the Fisher Energy associated with each direction. For a rank-one perturbation $Z = uv^\top$, substituting this form into equation 13 yields its corresponding Fisher Energy:

$$\mathcal{E}(Z) = \operatorname{vec}(Z)^\top (S_X \otimes S_Y) \operatorname{vec}(Z)$$
$$= \left(v^\top S_X v\right)\left(u^\top S_Y u\right). \quad (14)$$

Hence, the characterization of a direction is determined by pairs $(u, v)$ that reflect the variance structure of both $S_Y$ and $S_X$. To make these subspaces operational within our framework, we leverage the pre-trained weight $W_0$ to construct tractable bases. Specifically, we form orthonormal surrogate bases by normalizing the Gram matrices of $W_0$:

$$\hat{V} = \operatorname{normalize}(W_0^\top W_0),$$
$$\hat{U} = \operatorname{normalize}(W_0 \hat{V}). \quad (15)$$

Each column pair $(\hat{U}[:, i], \hat{V}[:, i])$ defines a candidate direction, as justified in Appendix A.5. Based on equation 14,

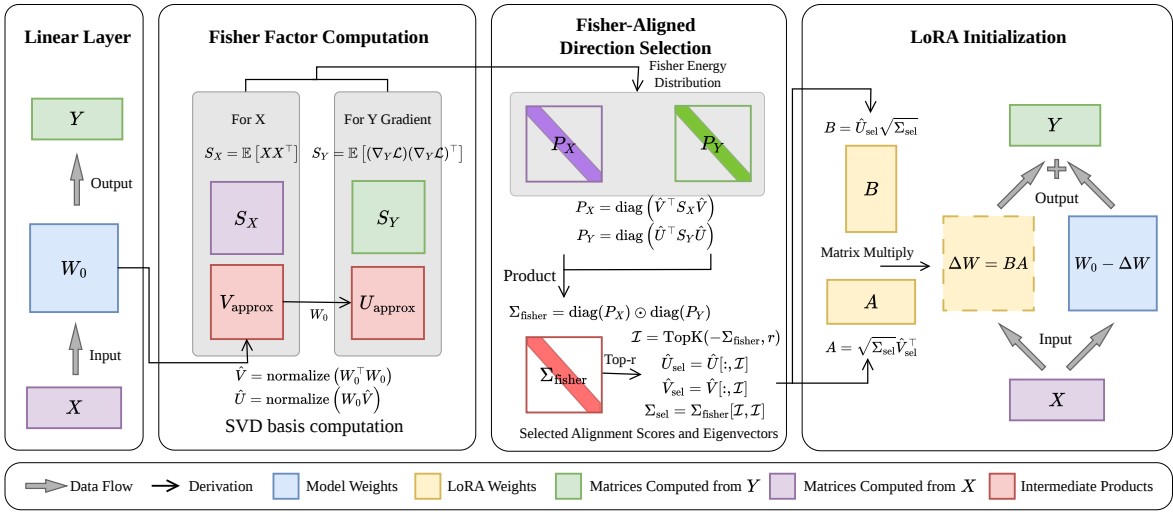

*Figure 2.* Overview of the proposed Fisher-Guided LoRA Initialization framework. The three subfigures correspond to its key components: **(a) Fisher Factor Computation**, where we compute the Fisher information using Kronecker-factored statistics using a minibatch of data; **(b) Fisher-Aligned Direction Selection**, where we identify Fisher-aligned directions by projecting onto surrogate bases derived from pre-trained weights; and **(c) LoRA Initialization**, where we compute weighted projections to initialize the LoRA factors.

we then compute the Fisher Energies by:

$$P_X = \hat{V}^\top S_X \hat{V},$$
$$P_Y = \hat{U}^\top S_Y \hat{U},$$
$$\Sigma_{\text{fisher}} = \text{diag}(P_X) \odot \text{diag}(P_Y), \qquad (16)$$

where $\Sigma_{\text{fisher}}$ is a diagonal matrix containing the Fisher Energies of all candidate directions. The top-$r$ directions with minimal $\Sigma_{\text{fisher}}$ are selected to form the Fisher-aligned subspaces:

$$\mathcal{I} = \text{TopK}(-\Sigma_{\text{fisher}}, r),$$
$$\hat{V}_{\text{sel}} = \hat{V}[:, \mathcal{I}],$$
$$\hat{U}_{\text{sel}} = \hat{U}[:, \mathcal{I}],$$
$$\Sigma_{\text{sel}} = \Sigma_{\text{fisher}}[\mathcal{I}, \mathcal{I}]. \qquad (17)$$

Given the Fisher-aligned subspaces $(\hat{U}_{\text{sel}}, \hat{V}_{\text{sel}})$, we compute the LoRA factors:

$$A = \sqrt{\Sigma_{\text{sel}}} \hat{V}_{\text{sel}}^\top,$$
$$B = \hat{U}_{\text{sel}} \sqrt{\Sigma_{\text{sel}}}, \qquad (18)$$

which satisfy $BA = \hat{U}_{\text{sel}} \Sigma_{\text{sel}} \hat{V}_{\text{sel}}^\top$. In this way, the weight update $\Delta W = \alpha(BA)$ lies in the Fisher-aligned subspace, with magnitudes scaled by the corresponding Fisher energies. Therefore, the pre-trained weight is decomposed as

$$W_0 = W_{\text{res}} + \alpha(BA), \qquad (19)$$

where $W_{\text{res}}$ retains the residual component orthogonal to the Fisher-aligned subspace.

## 4. Experiments

We conduct a series of experiments to assess the effectiveness of the proposed FILet method in improving LoRA fine-tuning performance. The experiments are structured into seven components, with *4.1 Main Experiments* focusing on evaluating performance across natural language reasoning benchmarks and image classification datasets; *4.2 Experiments on Natural Language Generation Tasks* examining the method's impact on text generation quality; *4.3 Rank Experiments* analyzing the rank of the learned low-rank adapters; *4.4 Initialization Time Analysis* comparing the computational efficiency of FILet against baseline methods; and *4.5 Ablation Study* investigating the contributions of key components within the FILet framework.

Comprehensive introductions of baseline methods are provided in Appendix B and all hyperparameters are detailed in Appendix C. In particular, we select KaSA as a representative SVD-based LoRA initialization baseline and LoRA-Dash as a representative data-driven LoRA enhancement baseline for comparison, as both methods have demonstrated strong performance across a wide range of tasks. For all experiments, we set the minibatch size to 320.

### 4.1. Main Experiments

For natural language reasoning tasks, we evaluate the proposed FILet on a suite of diverse reasoning benchmarks, including BoolQ (Clark et al., 2019), PIQA (Bisk et al., 2019), SIQA (Sap et al., 2019), HellaSwag (Zellers et al., 2019), WinoGrande (Sakaguchi et al., 2019), ARC-easy, ARC-challenge (Clark et al., 2018), and OBQA (Mihaylov

*Table 1.* Experimental results on natural language reasoning tasks. Results marked with $^\dagger$ are taken from their respective original papers (Liu et al., 2024; Meng et al., 2024; Wang et al., 2025b; Zhong et al., 2025; Wang et al., 2025a; Si et al., 2025). All baselines use the same number of trainable parameters, and all experiments are conducted in BF16 precision. We report the median over five runs, with standard deviations shown as underlined values. The best result for each dataset is highlighted in **bold**.

| Model | PEFT | BoolQ | PIQA | SIQA | HellaS. | WinoG. | ARC-e | ARC-c | OBQA | Avg. |
|---|---|---|---|---|---|---|---|---|---|---|
| ChatGPT$^\dagger$ | - | 73.1 | 85.4 | 68.5 | 78.5 | 66.1 | 89.8 | 79.9 | 74.8 | 77.0 |
| Llama2-7b | LoRA$^\dagger$ | 69.8 | 79.9 | 79.5 | 83.6 | 82.6 | 79.8 | 64.7 | 81.0 | 77.6 |
| | DoRA$^\dagger$ | 71.8 | 83.7 | 76.0 | 89.1 | 82.6 | 83.7 | 68.2 | 82.4 | 79.7 |
| | PiSSA$^\dagger$ | 67.6 | 78.1 | 78.4 | 76.6 | 78.0 | 75.8 | 60.2 | 75.6 | 73.8 |
| | MiLoRA$^\dagger$ | 67.6 | 83.8 | 80.1 | 88.2 | 82.0 | 82.8 | 68.8 | 80.6 | 79.2 |
| | NEAT$^\dagger$ | 71.9 | 84.0 | 80.4 | 88.9 | 84.6 | 86.5 | **71.6** | **83.0** | 81.4 |
| | KaSA | $74.3_{0.71}$ | $85.2_{0.66}$ | $81.9_{0.32}$ | $93.4_{0.23}$ | $85.3_{0.59}$ | $86.3_{0.62}$ | $67.4_{0.87}$ | $80.2_{0.77}$ | $81.7_{0.22}$ |
| | LoRA-Dash | $73.9_{0.58}$ | $85.0_{0.79}$ | $82.0_{0.38}$ | **94.2**$_{0.19}$ | $84.3_{0.90}$ | $85.7_{0.49}$ | $68.6_{0.92}$ | $79.4_{0.79}$ | $81.7_{0.38}$ |
| | FILet | **75.0**$_{0.40}$ | **86.6**$_{0.50}$ | **82.8**$_{0.50}$ | $94.2_{0.22}$ | **85.7**$_{0.53}$ | **87.1**$_{0.58}$ | $70.5_{0.45}$ | $82.6_{0.52}$ | **83.1**$_{0.26}$ |
| Llama3-8b | LoRA$^\dagger$ | 70.8 | 85.2 | 79.9 | 91.7 | 84.3 | 84.2 | 71.2 | 79.0 | 80.8 |
| | DoRA$^\dagger$ | 71.8 | 83.7 | 76.0 | 89.1 | 82.6 | 83.7 | 68.2 | 82.4 | 79.7 |
| | PiSSA$^\dagger$ | 67.1 | 81.1 | 77.2 | 83.6 | 78.9 | 77.7 | 63.2 | 74.6 | 75.4 |
| | MiLoRA$^\dagger$ | 68.8 | 86.7 | 77.2 | 92.9 | 85.6 | 86.8 | 75.5 | 81.8 | 81.9 |
| | NEAT$^\dagger$ | 72.1 | 87.0 | 80.9 | 94.3 | 86.7 | 91.4 | 78.9 | 84.8 | 84.5 |
| | KaSA | $74.3_{0.71}$ | $87.8_{0.70}$ | $82.2_{0.55}$ | $95.2_{0.38}$ | $87.5_{0.54}$ | $91.6_{0.57}$ | **80.3**$_{0.63}$ | $87.0_{0.62}$ | $85.7_{0.23}$ |
| | LoRA-Dash | $74.7_{0.67}$ | $87.4_{0.53}$ | $82.0_{0.42}$ | $95.8_{0.30}$ | $87.2_{0.55}$ | $92.0_{0.43}$ | $80.0_{0.66}$ | $86.0_{1.06}$ | $85.6_{0.29}$ |
| | FILet | **75.4**$_{0.55}$ | **90.6**$_{0.57}$ | **82.8**$_{0.52}$ | **96.1**$_{0.28}$ | **89.0**$_{0.45}$ | **92.2**$_{0.58}$ | $79.9_{0.39}$ | **87.4**$_{0.61}$ | **86.7**$_{0.13}$ |

*Table 2.* Experimental results on image classification tasks using the ViT-B/32 model. All baselines use the same number of trainable parameters, and all experiments are conducted in BF16 precision. We report the median over five runs, with standard deviations shown as underlined values. The best-performing result for each dataset is highlighted in **bold**.

| PEFT | Cars | DTD | EuroSAT | GTSRB | RESISC45 | SUN397 | SVHN | Avg. |
|---|---|---|---|---|---|---|---|---|
| LoRA | $78.4_{0.80}$ | $69.1_{0.30}$ | $89.2_{0.24}$ | $51.4_{0.34}$ | $76.2_{0.20}$ | $75.8_{0.12}$ | $38.9_{0.70}$ | $68.4_{0.25}$ |
| DoRA | $77.3_{0.49}$ | $69.0_{0.31}$ | $89.4_{0.30}$ | **52.5**$_{0.49}$ | $76.3_{0.14}$ | $76.0_{0.12}$ | $40.2_{1.08}$ | $68.7_{0.31}$ |
| PiSSA | $77.1_{1.00}$ | $68.9_{0.29}$ | $89.2_{0.29}$ | $51.2_{0.36}$ | $76.1_{0.21}$ | $75.6_{0.11}$ | $40.1_{0.97}$ | $68.3_{0.27}$ |
| MiLoRA | $77.9_{0.61}$ | $68.7_{0.26}$ | $89.5_{0.34}$ | $51.0_{0.39}$ | $76.4_{0.13}$ | **76.2**$_{0.10}$ | $40.2_{1.14}$ | $68.6_{0.34}$ |
| KaSA | $78.2_{0.64}$ | $68.4_{0.38}$ | $89.1_{0.24}$ | $51.5_{0.34}$ | $76.3_{0.10}$ | $75.9_{0.15}$ | $40.9_{0.79}$ | $68.6_{0.33}$ |
| LoRA-Dash | $77.4_{0.66}$ | $68.1_{0.24}$ | **89.6**$_{0.25}$ | $51.1_{0.50}$ | **76.5**$_{0.06}$ | $76.1_{0.14}$ | $40.5_{1.15}$ | $68.5_{0.28}$ |
| FILet | **79.2**$_{0.62}$ | **69.5**$_{0.17}$ | $89.5_{0.27}$ | **52.5**$_{0.31}$ | **76.5**$_{0.10}$ | $76.1_{0.12}$ | **41.6**$_{0.84}$ | **69.3**$_{0.31}$ |

et al., 2018). We compare our approach with several baseline LoRA initialization strategies, encompassing both standard random initialization and data-aware variants. All experiments are conducted using two widely adopted large language models: Llama2-7B (Touvron et al., 2023) and Llama3-8B (Grattafiori et al., 2024). For most reasoning datasets, FILet consistently outperforms all baseline approaches on both Llama2-7B and Llama3-8B models, as shown in Table 1. Notably, FILet achieves substantial gains over SVD-based methods such as MiLoRA and KaSA, as well as the data-driven LoRA-Dash, demonstrating the advantages of incorporating Fisher-guided decomposition for direction selection and initialization.

For image classification tasks, we evaluate FILet on seven standard datasets: Cars (Krause et al., 2013), DTD (Cimpoi et al., 2014), EuroSAT (Helber et al., 2018), GTSRB (Haloi, 2016), RESISC45 (Cheng et al., 2017), SUN397 (Xiao et al., 2010), and SVHN (Netzer et al.). We utilize the ViT-B/32 (Dosovitskiy et al., 2021) model as our backbone and com-

pare FILet against the baseline LoRA initialization methods. The results in Table 2 show that FILet performs strongly across these datasets, with particularly notable gains on SVHN and Cars. FILet attains the highest average accuracy among all methods, demonstrating its effectiveness in improving LoRA fine-tuning for image classification tasks.

### 4.2. Experiments on Natural Language Generation Tasks

For natural language generation tasks, we assess FILet on several benchmarks, including GSM8k (Cobbe et al., 2021) for mathematical reasoning, MBPP (Austin et al., 2021) for code generation, E2E (Novikova et al., 2017) for data-to-text generation, and MT-bench (Zheng et al., 2023) for open-ended text generation. We conduct experiments using several different LLMs, including Llama2-7B (Touvron et al., 2023), Llama3-8B (Grattafiori et al., 2024), Gemma-7B (Team et al., 2024), and Qwen2.5-7B (Qwen et al., 2025).

*Table 3*. Experimental results on natural language generation tasks. All experiments are conducted with BF16 precision. The best-performing result for each dataset is highlighted in **bold**.

| Model | PEFT | GSM8k | MBPP | BLEU | NIST | E2E METEOR | Rouge-L | CIDEr | MT-bench |
|---|---|---|---|---|---|---|---|---|---|
| Llama2-7b | LoRA | 32.9 | 15.7 | 39.8 | 5.50 | 65.6 | 54.4 | 12.9 | 4.26 |
| | KaSA | 33.2 | 15.6 | 39.3 | 5.43 | 65.4 | 54.5 | 12.5 | 4.12 |
| | LoRA-Dash | 34.1 | 15.8 | 40.1 | 5.55 | 65.8 | 54.6 | 12.8 | 4.29 |
| | FILet | **34.5** | **16.6** | **40.7** | **5.57** | **66.5** | **54.9** | **13.3** | **4.39** |
| Llama3-8b | LoRA | 55.6 | 22.2 | 39.3 | 5.47 | 65.6 | 54.9 | 12.4 | 4.91 |
| | KaSA | 53.7 | 23.4 | 40.4 | 5.48 | 66.0 | 55.0 | 12.9 | 4.75 |
| | LoRA-Dash | 55.8 | 22.8 | 40.5 | **5.60** | **66.3** | 54.8 | **13.2** | 4.95 |
| | FILet | **58.8** | **27.1** | **40.7** | 5.56 | **66.3** | **55.1** | 13.1 | **5.01** |
| Gemma-7b | LoRA | 54.1 | 29.4 | 39.7 | 5.49 | 65.6 | 54.0 | 12.6 | 4.32 |
| | KaSA | 54.5 | 29.9 | 38.9 | 5.41 | **65.8** | 54.4 | 12.8 | 4.37 |
| | LoRA-Dash | 58.7 | 29.5 | 39.3 | 5.46 | 65.0 | 54.5 | 12.9 | 4.43 |
| | FILet | **63.5** | **30.8** | **39.9** | **5.48** | 65.4 | **54.7** | **13.0** | **4.56** |
| Qwen2.5-7b | LoRA | 77.6 | 39.3 | 42.2 | 5.64 | 66.6 | 55.7 | 13.5 | 4.99 |
| | KaSA | 76.4 | 40.2 | 41.8 | 5.63 | 66.7 | 55.6 | 13.3 | 5.05 |
| | LoRA-Dash | 75.9 | 39.2 | 42.3 | 5.65 | **67.4** | 56.1 | 13.5 | 5.03 |
| | FILet | **78.8** | **42.5** | **42.4** | **5.68** | 66.9 | **56.3** | **13.7** | **5.11** |

The results in Table 3 show that FILet consistently outperforms all baseline methods across the evaluated models and tasks. In particular, FILet achieves substantial gains on GSM8K and MBPP, two benchmarks that require strong and precise generalization to novel linguistic patterns and structured grammar. These improvements indicate that FILet is especially effective for tasks that demand the incorporation of additional domain-specific knowledge and reasoning capabilities during adaptation, with mathematical problem solving and code generation serving as representative examples.

### 4.3. Rank Experiments

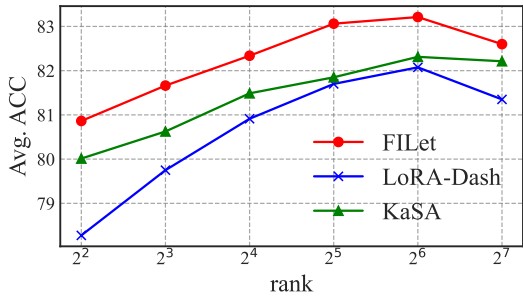

*Figure 3*. Experimental results of varying LoRA ranks on Llama2-7B. Average accuracy across reasoning tasks is reported.

We further investigate the impact of LoRA rank on fine-tuning performance. Figure 3 presents the average accuracy across reasoning tasks for varying LoRA ranks on Llama2-7B. The complete results of reasoning dataset are provided in Appendix D. The rank determines the capacity of the low-rank adapters and the number of Fisher-aligned directions selected during initialization. We evaluate ranks ranging from 4 to 128, spanning low- to high-rank settings, to assess the effectiveness of FILet under varying degrees of initialization capacity.

The results demonstrate that FILet maintains strong performance in low-dimensional regimes without suffering notable degradation. In particular, FILet achieves its best performance at rank 32 and 64, underscoring its ability to efficiently identify and exploit task-relevant adaptation directions to yield substantial performance gains. As the rank increases, FILet consistently outperforms baseline methods, further highlighting its effectiveness in selectively capturing and appropriately scaling the most beneficial directions for downstream tasks.

### 4.4. Computation Time

We present a comparison of additional initialization time relative to LoRA for FILet and the baseline methods on various sizes of LLMs in Figure 4.

For KaSA, LoRA-Dash, and other SVD-based methods, the initialization cost is dominated by performing SVD on pre-trained weight matrices, whereas for FILet it includes Fisher Factor Computation, Fisher-Aligned Direction Selection and LoRA Initialization. For SVD-based methods, weights must be cast to FP32 for numerical stability during SVD, which incurs substantial overhead for large-scale models. In contrast, FILet does not require exact singular bases; instead, it only needs surrogate singular bases $\tilde{U}$ and $\hat{V}$ together with empirical second-moment matrices $S_X$ and $S_Y$ from minibatch statistics. This design avoids full SVD

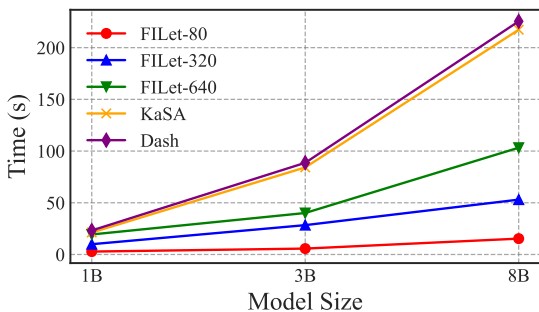

*Figure 4.* Extra initialization time comparison. We report the total additional initialization time (in seconds) for different LoRA initialization methods using an input length of 512, rank $r = 32$, and BF16 precision, measured on a single NVIDIA A100 GPU. Three model scales are evaluated: Llama3.2-1B ("1B"), Llama3.2-3B ("3B"), and Llama3-8B ("8B"). For KaSA, we report the full initialization time, whereas for LoRA-Dash we report only the additional time incurred between the pre-launch and dash phases.

computation as well as costly precision conversions, reducing the overall procedure to pure matrix multiplications. Consequently, FILet incurs substantially lower initialization overhead on LLMs, with most of the remaining time devoted to gradient computation. Notably, this efficiency advantage becomes increasingly pronounced as model size grows.

## 4.5. Ablation Study

For the ablation study, we investigate two key components of FILet: minibatch size for initialization and the initialization strategies.

### 4.5.1. MINIBATCH SIZE ANALYSIS

We study the effect of minibatch size on the estimation of the empirical second-moment matrices $S_X$ and $S_Y$ in FILet. Figure 5 reports the average accuracy across reasoning tasks for different minibatch sizes on Llama2-7B and Llama3-8B, with full results provided in Appendix E. We consider five minibatch sizes of 80, 160, 320, 480, and 640, and additionally include a "Full" setting that uses the entire dataset as a reference.

Overall, increasing the minibatch size leads to more accurate estimates of the second-moment matrices and correspondingly improves downstream performance. For instance, increasing the minibatch size from 80 to *Full* yields gains of up to 0.7% on Llama2-7B and 0.9% on Llama3-8B. However, these improvements exhibit clear diminishing returns: performance gains become marginal when increasing the minibatch size from 480 to 640 or when using the full dataset. This suggests that FILet can achieve strong performance with relatively moderate minibatch sizes, effectively balancing computational efficiency and estimation fidelity.

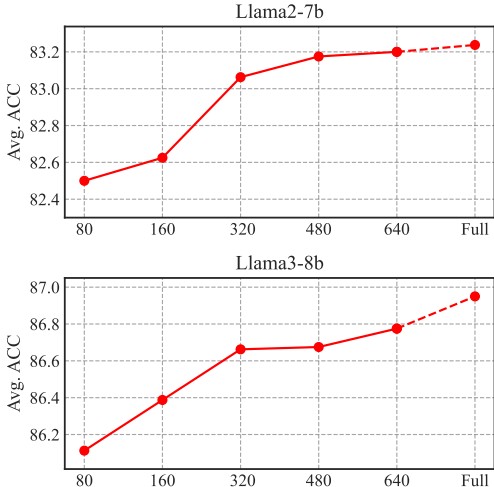

*Figure 5.* The average performance on reasoning tasks of the minibatch size experiments on the Llama2-7B and Llama3-8B models.

### 4.5.2. INITIALIZATION STRATEGY ANALYSIS

*Table 4.* Experimental results of different initialization strategies. FILet-SVD-P, FILet-SVD-R, and FILet-SVD-M denote variants that select directions with maximal, random, and minimal Fisher Energy, respectively, and scale them using the corresponding singular values. FILet-P, FILet-R, and FILet-M denote the corresponding variants that use Fisher-based scaling. All experiments are conducted with BF16 precision, and results are reported as the median over five runs. The best-performing result for each dataset is highlighted in **bold**.

|            | Llama2-7b | Llama3-8b | ViT-B/32 |
| :--------- | :-------: | :-------: | :------: |
| PEFT       | Reasoning | Reasoning | Image Cls. |
| FILet-SVD-P | 80.9 | 84.7 | 68.3 |
| FILet-P     | 81.8 | 85.8 | 68.4 |
| FILet-SVD-R | 81.3 | 84.8 | 68.6 |
| FILet-R     | 82.5 | 86.4 | 68.9 |
| FILet-SVD-M | 81.9 | 85.8 | 68.3 |
| FILet-M     | **83.1** | **86.7** | **69.3** |

We compare the proposed Fisher-based scaling strategy with alternative initialization approaches that use different direction-selection criteria and exact singular values for scaling, schemes across natural language reasoning and image classification tasks, using Llama2-7B, Llama3-8B, and ViT-B/32. The results are summarized in Table 4, with full results and detailed descriptions of all baselines provided in Appendix F.

If direction scaling is using the SVD singular values rather than Fisher Energies, the initialization directions are not properly scaled or aligned with the Fisher Energy landscape, and thus fail to reflect true parameter sensitivity. As a result, performance consistently degrades across all datasets. Conversely, even when Fisher-based scaling is applied, se-

lecting random or maximal Fisher energy directions, that is, directions that are poorly chosen despite being scaled under the Fisher metric, still leads to suboptimal performance compared with the proposed FILet method. These results underscores the importance of selecting directions with minimal Fisher Energy and scaling them accordingly in achieving effective LoRA initialization.

Additionally, to further elucidate the task specificity of the proposed direction selection mechanism, we provide a detailed analysis of the overlap among initialization directions across reasoning and generation tasks in Appendix G.

## 5. Conclusion

We presented FILet, a data-aware LoRA initialization method that leverages Fisher Energy to identify and scale adaptation directions that are most effective for downstream adaptation. By efficiently estimating empirical second-moment statistics from minibatches, FILet captures the local parameter sensitivity induced by the target data distribution without incurring significant computational overhead. Extensive experiments across natural language reasoning, natural language generation, and image classification tasks demonstrate that FILet consistently outperforms existing LoRA initialization strategies across a variety of large language models, achieving stronger downstream performance while maintaining minimal initialization time and computational complexity. Additionally, we present the limitation analysis in Appendix H.

## Impact Statement

PEFT methods such as LoRA have become popular for adapting large pre-trained models to downstream tasks due to their efficiency and effectiveness. Our proposed FILet method further enhances LoRA by introducing a principled, data-aware, and computationally efficient initialization strategy that consistently improves downstream task performance across diverse model architectures and application domains. By enabling more effective adaptation with minimal additional overhead, FILet strengthens the practical utility of PEFT techniques, allowing high-quality fine-tuning under limited computational budgets. While FILet does not expand the intrinsic capabilities of pre-trained models, its improved efficiency and performance may facilitate broader and more sustainable use of large models in real-world settings, including scenarios where computational resources are constrained.

## Acknowledgements

We would like to sincerely acknowledge the support from the National Science and Technology Council (NSTC), Taiwan, under Grant NSTC 115-2634-F-007-006, and the Qualcomm Innovation Fellowship (QIF) under Grant B114-K654.

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

# A. Mathematical Details

## A.1. Symbols and Notations

We summarize the notation used throughout this paper for clarity in Table 5. All vectors are column vectors unless otherwise stated, and all expectations are taken over the empirical data distribution.

*Table 5.* List of symbols and notations used in this paper.

| Type | Symbol | Dimension | Definition |
|---|---|---|---|
| Scalars | $n$ | – | Input dimension of the linear layer. |
| | $m$ | – | Output dimension of the linear layer. |
| | $l$ | – | Input sequence length (number of tokens/time-steps). |
| | $r$ | – | Target rank of the low-rank adaptation. |
| | $\alpha$ | – | LoRA rank scaling factor. |
| | scale | $\mathbb{R}$ | LoRA scaling coefficient controlling update magnitude. Given by scale $= \frac{\alpha}{r}$. |
| Matrices / Vectors | $X$ | $\mathbb{R}^{n \times 1}$ | Input activation for a mini-batch. |
| | $Y$ | $\mathbb{R}^{m \times 1}$ | Output activation: $Y = WX$. |
| | $W$ | $\mathbb{R}^{m \times n}$ | Weight matrix of the linear transformation. |
| | $W_0$ | $\mathbb{R}^{m \times n}$ | Pretrained weight matrix. |
| | $\Delta W$ | $\mathbb{R}^{m \times n}$ | Low-rank weight update: $\Delta W = \text{scale} \cdot (BA)$. |
| | $W_{\text{res}}$ | $\mathbb{R}^{m \times n}$ | Residual weight after removing Fisher-aligned components. |
| | $A$ | $\mathbb{R}^{r \times n}$ | LoRA input-side factor. |
| | $B$ | $\mathbb{R}^{m \times r}$ | LoRA output-side factor. |
| | $S_X$ | $\mathbb{R}^{n \times n}$ | Empirical input covariance (Fisher factor): $S_X = \mathbb{E}[XX^\top]$. |
| | $S_Y$ | $\mathbb{R}^{m \times m}$ | Empirical output gradient covariance (Fisher factor): $S_Y = \mathbb{E}[(\nabla_Y \mathcal{L})(\nabla_Y \mathcal{L})^\top]$. |
| | $\nabla_Y \mathcal{L}$ | $\mathbb{R}^{m \times 1}$ | Gradient of the loss w.r.t. the layer output. |
| | $G_X$ | $\mathbb{R}^{n \times n}$ | Right Gram matrix of pre-trained weights: $G_X = W_0^\top W_0$. |
| | $\hat{V}$ | $\mathbb{R}^{n \times n}$ | The surrogate right singular basis computed from $G_X$, column-wise normalized. |
| | $\hat{U}$ | $\mathbb{R}^{m \times n}$ | The surrogate left singular basis: $\hat{U} = \text{normalize}(W_0 \hat{V})$. |
| | $\hat{V}_{\text{sel}}$ | $\mathbb{R}^{n \times r}$ | Selected right-basis (Fisher-aligned columns from $\hat{V}$). |
| | $\hat{U}_{\text{sel}}$ | $\mathbb{R}^{m \times r}$ | Selected left-basis (Fisher-aligned columns from $\hat{U}$). |
| | $P_X$ | $\mathbb{R}^{m \times n}$ | Projected input Fisher: $P_X = \hat{V}^\top S_X \hat{V}$. |
| | $P_Y$ | $\mathbb{R}^{m \times n}$ | Projected output Fisher: $P_Y = \hat{U}^\top S_Y \hat{U}$. |
| | $U$ | $\mathbb{R}^{m \times m}$ | The left singular values of $W_0$. |
| | $V$ | $\mathbb{R}^{n \times n}$ | The right singular values of $W_0$. |
| | $\Sigma$ | $\mathbb{R}^{m \times n}$ | The singular values of $W_0$. |
| | $\Sigma_{\text{fisher}}$ | $\mathbb{R}^{n \times n}$ | The Fisher Energies of the candidate directions: $\Sigma_{\text{fisher}} = \text{diag}(P_X) \odot \text{diag}(P_Y)$. |
| | $\Sigma_{\text{sel}}$ | $\mathbb{R}^{r \times r}$ | The Fisher Energies associated with the selected $r$ directions (Fisher-aligned values from $\Sigma_{\text{fisher}}$). |
| Others / Operators | $\mathcal{L}$ | – | Task-specific loss function. |
| | $\text{normalize}(\cdot)$ | – | Column-wise normalization operator (each column to unit $\ell_2$ norm). |
| | $\text{diag}(\cdot)$ | – | Constructs a diagonal matrix from a vector. |
| | $\text{TopK}(\cdot, r)$ | – | Operator that returns indices of the top-$r$ values. |
| | $\mathcal{I}$ | – | The list of selected top-$r$ indices which have the lowest Fisher Energies. |
| | $\otimes$ | – | The Kronecker product. |

The table above summarizes the key symbols and notations used throughout this paper. Scalars, matrices, and operators are listed separately to provide clarity and facilitate consistent usage in the subsequent derivations and algorithm descriptions.

## A.2. Algorithm Overview

In this subsection, we provide a high-level overview of the proposed Fisher-Guided LoRA initialization procedure. The goal is to identify the directions in the pre-trained weight space that are most sensitive according to the Fisher information and to construct low-rank LoRA matrices aligned with these directions. This approach ensures that the initial adaptation focuses on the most informative components of the model, facilitating faster convergence and improved performance during fine-tuning.

Algorithm 1 efficiently extracts the most informative directions from the pre-trained weight space and constructs the corresponding low-rank adaptation matrices. By focusing updates along these Fisher-aligned directions, the method reduces unnecessary perturbations in less informative components and provides a strong initialization for subsequent fine-tuning on

---

**Algorithm 1** Fisher-Guided LoRA Initialization

---

**Input:** pre-trained weights $W_0 \in \mathbb{R}^{m \times n}$, data loader $\mathcal{D}$, rank $r$, scale $\alpha$
**Output:** LoRA matrices $A, B$ and residual weight $W_{\text{res}}$
Initialize covariance accumulators: $S_X \leftarrow 0, S_Y \leftarrow 0$
**repeat**
   Sample mini-batch $(X, Y) \sim \mathcal{D}$
   Compute model output $h = f_{W_0}(X)$ and loss $\mathcal{L}(h, Y)$
   Backpropagate to obtain gradients $\nabla_Y \mathcal{L}$ for each linear layer
   Record activations $X$ and gradients $\nabla_Y \mathcal{L}$
   $S_X \mathrel{+}= XX^\top / \dim(X)$
   $S_Y \mathrel{+}= (\nabla_Y \mathcal{L})(\nabla_Y \mathcal{L})^\top / \dim(\nabla_Y \mathcal{L})$
**until** All data in the mini-batche processed
Normalize: $S_X \leftarrow S_X / T, \quad S_Y \leftarrow S_Y / T$
Compute statistics of input and output gradients: $G_X = W_0^\top W_0$
Form the bases:
   $\hat{V} = \text{normalize}(G_X)$
   $\hat{U} = \text{normalize}(W_0 \hat{V})$
Projected Fisher correlations:
   $P_X = \hat{V}^\top S_X \hat{V}$
   $P_Y = \hat{U}^\top S_Y \hat{U}$
Compute Fisher Energies: $\Sigma_{\text{fisher}} = \text{diag}(P_X) \odot \text{diag}(P_Y)$
Select directions:
   $\mathcal{I} = \text{TopK}(-\Sigma_{\text{fisher}}, r)$
Form selected subspaces:
   $\hat{V}_{\text{sel}} = \hat{V}[:, \mathcal{I}]$
   $\hat{U}_{\text{sel}} = \hat{U}[:, \mathcal{I}]$
   $\Sigma_{\text{sel}} = \Sigma_{\text{fisher}}[\mathcal{I}, \mathcal{I}]$
Construct LoRA matrices:
   $A = \sqrt{\Sigma_{\text{sel}}} \hat{V}_{\text{sel}}^\top$
   $B = \hat{U}_{\text{sel}} \sqrt{\Sigma_{\text{sel}}}$
Compute residual: $W_{\text{res}} = W_0 - \alpha(BA)$
**Return:** $A, B, W_{\text{res}}$

---

downstream tasks.

## A.3. Full Fisher Information Matrix: Definition, Structure, and Infeasibility

This section provides a detailed characterization of the full Fisher information matrix associated with a linear transformation layer $(Y = WX)$, where the notations follow the symbol table: $X \in \mathbb{R}^{n \times 1}$, $Y \in \mathbb{R}^{m \times 1}$, and $W \in \mathbb{R}^{m \times n}$. We describe the mathematical and geometric interpretation of each quantity, the resulting matrix shape, and why computing or eigendecomposing the full Fisher is **infeasible** for modern large models.

For a weight matrix $W \in \mathbb{R}^{m \times n}$, the Fisher information matrix is defined as

$$F = \mathbb{E}\left[gg^\top\right], \qquad g = \text{vec}\left(\nabla_W \mathcal{L}\right). \tag{20}$$

Let $W_{ij}$ and $W_{kl}$ be two coordinates of the weight matrix. Then the entry $((i,j),(k,l))$ of the Fisher is

$$F_{(i,j),(k,l)} = \mathbb{E}\left[\frac{\partial \log p(x \mid W)}{\partial W_{ij}} \cdot \frac{\partial \log p(x \mid W)}{\partial W_{kl}}\right]. \tag{21}$$

Computing the full Fisher requires:

1. Computing the full gradient $g \in \mathbb{R}^{mn}$ for each training example.

2. Forming the outer product $gg^\top \in \mathbb{R}^{(mn)\times(mn)}$.

3. Averaging these outer products over the dataset.

For example, if $W$ is a projection layer with $m = 4096$ and $n = 4096$, then $mn \approx 1.68 \times 10^7$, and $F$ contains $(mn)^2 \approx 2.8 \times 10^{14}$ entries. Even the storage requirement is already prohibitive:

$$(mn)^2 \text{ entries } \times 4 \text{ bytes} \gg 1 \text{ TB} \quad \text{for modern layer sizes.}$$

Large Language Models (LLMs) with billions of parameters would require storing matrices with up to $10^{18}$–$10^{22}$ entries—far beyond the memory capacity of any modern GPU cluster. Thus, instead of computing the full Fisher, we seek efficient factorization that capture its essential structure without incurring prohibitive computational costs.

### A.4. Proof of Fisher Factorization

**Lemma A.1. Gradient Factorization Identity**
*For a linear layer defined as $Y = WX$, the gradient with respect to the weight matrix $W$ admits the following factorized form:*

$$\nabla_W \mathcal{L} = (\nabla_Y \mathcal{L})X^\top.$$

*Proof.* Consider a linear transformation in matrix form:

$$Y = WX, \tag{22}$$

where $W \in \mathbb{R}^{m\times n}$, $X \in \mathbb{R}^{n\times 1}$, and $Y \in \mathbb{R}^{m\times 1}$. Let $\mathcal{L}(Y) : \mathbb{R}^{m\times 1} \to \mathbb{R}$ be a scalar loss function. We compute the total differential of $\mathcal{L}$ induced by a small perturbation $dW$ of $W$. The differential of the layer output is

$$dY = d(WX) = (dW)X, \tag{23}$$

since $X$ is constant during differentiation.

The differential of the scalar loss can be expressed using the Frobenius inner product:

$$
\begin{aligned}
d\mathcal{L} &= \big\langle \nabla_Y \mathcal{L}, dY \big\rangle_F \\
&= \big\langle \nabla_Y \mathcal{L}, (dW)X \big\rangle_F \\
&= \text{trace}\left( (\nabla_Y \mathcal{L})^\top (dW)X \right) \\
&= \big\langle (\nabla_Y \mathcal{L})X^\top, dW \big\rangle_F .
\end{aligned}
\tag{24}
$$

By the definition of the gradient with respect to the Frobenius inner product, the unique matrix satisfying $d\mathcal{L} = \langle \nabla_W \mathcal{L}, dW \rangle_F$ for all perturbations $dW$ is

$$\nabla_W \mathcal{L} = (\nabla_Y \mathcal{L})X^\top. \tag{25}$$

This precisely matches Lemma A.1. □

## A.5. Justification of Surrogate Basis Construction

> **Lemma A.2. Construction of Surrogate Singular Basis**
>
> *Let $W_0 \in \mathbb{R}^{m \times n}$ denote the pre-trained weight matrix. Define its Gram matrices:*
>
> $$G_X = W_0^\top W_0 \in \mathbb{R}^{n \times n}.$$
>
> *We construct the surrogate singular bases $\hat{V}$ and $\hat{U}$ as:*
>
> $$\hat{V} = \text{normalize}(G_X), \qquad \hat{U} = \text{normalize}(W_0 \hat{V}),$$
>
> *where* normalize($\cdot$) *denotes column-wise $\ell_2$ normalization. While distinct from the exact SVD, each column pair $(\hat{U}[:,i], \hat{V}[:,i])$ lies within the active singular subspaces of $W_0$. Due to the spectral properties of $G_X$, these vectors serve as a computationally tractable proxy for the principal components of the Fisher eigenbasis.*

Consider the full SVD of $W_0$:

$$W_0 = U\Sigma V^\top, \tag{26}$$

where $U \in \mathbb{R}^{m \times m}$ and $V \in \mathbb{R}^{n \times n}$ are orthogonal matrices containing the true left and right singular vectors, and $\Sigma \in \mathbb{R}^{m \times n}$ contains the singular values $\sigma_1 \geq \sigma_2 \geq \cdots \geq 0$.

Using this decomposition, we express the right Gram matrix $G_X$:

$$G_X = W_0^\top W_0 = (U\Sigma V^\top)^\top (U\Sigma V^\top) = V\Sigma^\top U^\top U\Sigma V^\top = V\Sigma^2 V^\top, \tag{27}$$

where $\Sigma^2 = \text{diag}(\sigma_1^2, \ldots, \sigma_n^2)$. Equation equation 27 reveals that the columns of $G_X$ are linear combinations of the true right singular vectors $V$, weighted by the squared singular values $\sigma_i^2$.

The column space (range) of $G_X$ is identical to the subspace spanned by the right singular vectors corresponding to non-zero singular values:

$$\text{range}(G_X) = \text{span}\{v_i \mid \sigma_i > 0\} \subseteq \text{range}(V). \tag{28}$$

We define our surrogate input basis $\hat{V}$ by column-wise normalization:

$$\hat{V} := \text{normalize}(G_X). \tag{29}$$

Let $g_j$ be the $j$-th column of $G_X$. Based on the spectral expansion, $g_j = \sum_k (\sigma_k^2 V_{jk}) v_k$. Normalizing $g_j$ to obtain $\hat{v}_j$ preserves its direction. Consequently, every column $\hat{v}_j$ of $\hat{V}$ remains strictly within the active right singular subspace of $W_0$. Furthermore, due to the weighting factor $\sigma_k^2$, these directions are heavily biased towards the principal singular vectors (those with large $\sigma_k$), making $\hat{V}$ an effective heuristic for capturing the dominant input sensitivities.

Given the surrogate input directions $\hat{V}$, we map them to the output space via $W_0$:

$$W_0 \hat{V} = (U\Sigma V^\top)\hat{V}. \tag{30}$$

Since each column $\hat{v}_j$ lies in the span of the right singular vectors $V$, the linear transformation $W_0$ maps it directly into the subspace spanned by the left singular vectors $U$. Specifically, if $\hat{v}_j = \sum_k \alpha_k v_k$, then $W_0 \hat{v}_j = \sum_k \alpha_k \sigma_k u_k$. This implies that the columns of $W_0 \hat{V}$ lie strictly within the active left singular subspace of $W_0$.

To ensure consistency in magnitude, we compute the surrogate output basis $\hat{U}$ via normalization:

$$\hat{U} := \text{normalize}(W_0 \hat{V}). \tag{31}$$

Thus, the constructed pairs $(\hat{u}_j, \hat{v}_j)$ are valid directions within the left and right singular subspaces of $W_0$, respectively. While $\hat{V}$ and $\hat{U}$ may not be strictly orthogonal like the true SVD bases, they provide a computationally efficient approximation that aligns with the data-driven mapping induced by the pre-trained weights.

## B. Baselines

Here are the baselines we compare against in our experiments:

- **LoRA** (Hu et al., 2022): The original Low-Rank Adaptation method that adds trainable low-rank matrices to each weight matrix. LoRA freezes the pre-trained weights and injects a low-rank update $\Delta W = BA$ during fine-tuning.

- **DoRA** (Liu et al., 2024): DoRA normalizes the weight direction and magnitude separately, applying low-rank adaptation only to the direction component.

- **PiSSA** (Meng et al., 2024): PiSSA applies SVD to the pre-trained weights and utilizes the principal singular values and their corresponding singular vectors to initialize the LoRA module, while retaining the remaining components as fixed bases for adaptation.

- **MiLoRA** (Wang et al., 2025b): MiLoRA focuses on the least-dominant singular subspace that is underutilized during pre-training and uses them to initialize the LoRA module while using the others as fixed bases. It adapts directions orthogonal to the dominant subspace, promoting better task specialization and mitigating interference from pre-trained knowledge.

- **LoRA-Dash** (Si et al., 2025): LoRA-Dash adopts a two-stage adaptation strategy. In the first stage, standard LoRA fine-tuning is conducted to identify the most relevant adaptation directions. In the second stage, these learned directions are reused to project the weights onto more suitable subspaces for effective adaptation.

- **NEAT** (Zhong et al., 2025): NEAT introduces a nonlinear adaptation mechanism that models updates as a function of pre-trained weights. This nonlinear formulation enables NEAT to capture more complex parameter interactions while preserving parameter efficiency.

- **KaSA** (Wang et al., 2025a): KASA leverages SVD to enable knowledge-aware and structured parameter adaptation. It first truncates unimportant singular components to preserve essential world knowledge, then applies task-specific updates in the SVD space.

## C. Hyperparameters

In this section, we detail the hyperparameter configurations and prompting strategies used throughout our experiments. All experiments are conducted on a single NVIDIA A100 GPU with 80GB of memory, with BF16 precision enabled to ensure computational efficiency while maintaining numerical stability during both training and inference.

*Table 6.* Hyperparameters for reasoning tasks. In reasoning experiments, we use the same hyperparameters on both Llama2-7B and Llama3-8B models.

| | BoolQ | PIQA | SIQA | HellaS. | WinoG. | ARC-e | ARC-c | OBQA |
|---|---|---|---|---|---|---|---|---|
| rank | | | | 3 | 2 | | | |
| alpha | 64 | 64 | 64 | 64 | 64 | 32 | 32 | 64 |
| optimizer | | | | Ada | mW | | | |
| learning rate | 1e-4 | 1e-4 | 1e-4 | 8e-5 | 9e-5 | 1e-4 | 1e-4 | 1e-4 |
| batch size | | | | 4 | | | | |
| epochs | 3 | 4 | 3 | 3 | 3 | 6 | 10 | 10 |
| warmup steps | 200 | 250 | 400 | 1000 | 500 | 100 | 100 | 500 |
| dropout | | | | 0.0 | 5 | | | |
| modules | | | | q, k, v, u | p, down | | | |

For the main experiments reported in Section 4.1, we adopt the hyperparameters summarized in Table 6 for natural language reasoning tasks and in Table 8 for image classification tasks. The corresponding prompt templates used for the reasoning benchmarks are provided in Table 7.

In the rank experiments described in Section 4.3, we adjust the optimization hyperparameters according to the selected LoRA rank. Specifically, we employ larger learning rates for lower-rank settings and progressively smaller learning rates as the rank increases, in order to maintain training stability and mitigate overfitting. For the initialization time experiments in Section 4.4, we apply LoRA initialization to the same set of modules as in the reasoning experiments. For all ablation

*Table 7.* Prompts used for reasoning tasks. The tokens [Question], [Context], [Sentence], [Choice A], [Choice B], and [Choice C] denote placeholders that are instantiated with the specific fields of each dataset example during evaluation.

| Dataset | Prompt |
|---|---|
| **BoolQ** | Decide if the question can be answered with True or False. Question: [Question] Answer: |
| **PIQA** | Choose the solution that best achieves the goal. Question: [Question] Choices: A:[Choice A] B:[Choice B] Answer: |
| **SIQA** | Answer the question based on the provided context and given choices. Context: [Context] Choices: A:[Choice A] B:[Choice B] ... Answer: |
| **HellaS.** | Choose the most appropriate ending for the given context. Context: [Context] Choices: A:[Choice A] B:[Choice B] ... Answer: |
| **WinoG.** | Choose the correct option that completes the sentence. Sentence: [Sentence] Choices: A:[Choice A] B:[Choice B] Answer: |
| **ARC-e** | Answer the question based on the given choices. Question: [Question] Choices: A:[Choice A] B:[Choice B] ... Answer: |
| **ARC-c** | Answer the question based on the given choices. Question: [Question] Choices: A:[Choice A] B:[Choice B] ... Answer: |
| **OBQA** | Please choose the correct answer to the question. Question: [Question] Choices: A:[Choice A] B:[Choice B] C:[Choice C] Answer: |

studies in Section 4.5, we keep the hyperparameter configurations identical to those used in the main experiments to enable fair and controlled comparisons.

*Table 8.* Hyperparameters for image classification tasks.

|  | **Cars** | **DTD** | **EuroSAT** | **GTSRB** | **RESISC45** | **SUN397** | **SVHN** |
|---|---|---|---|---|---|---|---|
| rank |  |  |  | 8 |  |  |  |
| alpha |  |  |  | 16 |  |  |  |
| learning rate |  |  |  | 2e-4 |  |  |  |
| batch size |  |  |  | 64 |  |  |  |
| epochs | 40 | 80 | 15 | 15 | 15 | 15 | 5 |
| optimizer |  |  |  | AdamW |  |  |  |
| warmup steps | 100 | 200 | 200 | 200 | 200 | 200 | 200 |
| dropout |  |  |  | 0.05 |  |  |  |
| modules |  |  |  | q, k, v |  |  |  |

For natural language generation tasks in section 4.2, we follow the hyperparameter settings in Table 9. The prompts used for natural language generation tasks are listed in Table 10. GSM8K and E2E are trained on their training sets, while MBPP is trained on CodeFeedBack (Weyssow et al., 2025) and MT-bench is trained on the training set of Alpaca (Taori et al., 2023).

*Table 9.* Hyperparameters for natural language generation tasks. We apply different learning rates for different base models while keeping other hyperparameters consistent across all models.

|  | **GSM8K** | **MBPP** | **E2E** | **MT-bench** |
|---|---|---|---|---|
| rank |  | 8 |  |  |
| alpha |  | 32 |  |  |
| batch size |  | 2 |  |  |
| epochs |  | 1 |  |  |
| optimizer |  | AdamW |  |  |
| warmup steps |  | 1000 |  |  |
| dropout |  | 0.05 |  |  |
| modules |  | q, k, v, up, down |  |  |
| learning rate (Llama2-7b) |  | 1.2e-4 |  |  |
| learning rate (Llama3-8b) |  | 8e-5 |  |  |
| learning rate (Gemma-7b) |  | 1.5e-5 |  |  |
| learning rate (Qwen2.5-7b) |  | 1.5e-5 |  |  |

*Table 10.* Prompts used for natural language generation tasks. [Input], [Context] and [Question] are placeholders to be replaced with the actual content from each dataset instance.

| Dataset | Prompt |
|---|---|
| GSM8K | Solve the grade-school math word problem step by step and provide the final numeric answer. [Input] Solution: |
| MBPP | Write a correct and efficient Python function that solves the given programming problem according to the specification. \n\n### Programming problem: [Inputs] \n\n### Code: |
| E2E | Generate a natural, varied, and coherent restaurant-domain description from the given meaning representation, reflecting appropriate content selection and rich linguistic expression. [Inputs] Answer: |
| MT-bench | Below is an instruction that describes a task. Write a response that appropriately completes the request. \n\n### Instruction: [Inputs] \n\n### Response: |

# D. Full Results of Section 4.3

In this section, we provide the complete results for the rank ablation experiments presented in Section 4.3. Table 11 summarizes the performance of FILet, LoRA-Dash, and KaSA across various ranks on the reasoning benchmarks.

Across these results, FILet consistently outperforms the baseline methods at nearly all rank settings and across all datasets, demonstrating both its robustness and its effectiveness in exploiting Fisher-aligned subspaces for low-rank adaptation. Notably, we observe that FILet and the competing baselines generally achieve their strongest performance at ranks 32 and 64, suggesting that this range offers a favorable trade-off between adaptation capacity and parameter efficiency for the reasoning tasks considered.

*Table 11.* Full experimental results under different ranks $r$ on reasoning benchmarks using Llama2-7B as the base model. The best results for each rank are highlighted in **bold**.

| PEFT | | BoolQ | PIQA | SIQA | HellaS. | WinoG. | ARC-e | ARC-c | OBQA | Avg. |
|---|---|---|---|---|---|---|---|---|---|---|
| FILet | $r = 4$ | 73.9 | 83.7 | 81.4 | 93.0 | 83.5 | 84.2 | 67.1 | 79.8 | 80.8 |
| | $r = 8$ | 74.4 | 84.7 | 82.1 | 93.4 | 84.1 | 85.0 | 68.5 | 80.8 | 81.6 |
| | $r = 16$ | 74.7 | 86.1 | 82.6 | 93.5 | 84.7 | 86.6 | 68.2 | 81.6 | 82.3 |
| | $r = 32$ | 75.0 | 86.6 | 82.8 | 94.2 | 85.7 | 87.1 | **70.5** | **82.6** | 83.1 |
| | $r = 64$ | **75.4** | **86.8** | **83.0** | **94.3** | **86.1** | **87.3** | 70.2 | **82.6** | **83.2** |
| | $r = 128$ | 74.8 | 85.9 | 82.4 | 94.0 | 85.9 | 86.8 | 69.0 | 82.0 | 82.6 |
| LoRA-Dash | $r = 4$ | 71.9 | 82.1 | 81.1 | 91.5 | 81.6 | 84.1 | 63.7 | 70.2 | 78.3 |
| | $r = 8$ | 72.4 | 82.5 | 81.5 | 92.0 | 83.1 | 85.1 | 67.8 | 73.6 | 79.8 |
| | $r = 16$ | 73.2 | 84.8 | 82.4 | 92.8 | 83.4 | 85.3 | 67.4 | 78.0 | 80.9 |
| | $r = 32$ | 73.9 | 85.0 | 82.5 | **94.2** | 84.3 | 85.7 | **68.6** | 79.4 | 81.7 |
| | $r = 64$ | **74.2** | **85.3** | **83.2** | 93.8 | **84.9** | **86.3** | 68.3 | **80.6** | **82.1** |
| | $r = 128$ | 73.9 | 84.8 | 82.3 | 93.6 | 84.6 | 85.3 | 67.3 | 79.0 | 81.4 |
| KaSA | $r = 4$ | 72.1 | 82.5 | 81.4 | 92.0 | 83.6 | 84.7 | 65.4 | 78.4 | 80.0 |
| | $r = 8$ | 73.4 | 83.7 | 81.6 | 92.5 | 84.2 | 85.7 | 65.1 | 78.8 | 80.6 |
| | $r = 16$ | 74.0 | 84.8 | 82.4 | 93.3 | 84.6 | 86.0 | 67.0 | 79.8 | 81.5 |
| | $r = 32$ | 74.3 | 85.2 | 81.9 | 93.4 | 86.1 | 86.3 | 67.4 | 80.2 | 81.8 |
| | $r = 64$ | **74.4** | 85.5 | **82.7** | **93.9** | **86.4** | 86.6 | **68.4** | 80.6 | **82.3** |
| | $r = 128$ | 74.1 | **85.6** | **82.7** | 93.8 | 85.6 | **87.0** | 67.3 | **81.6** | 82.2 |

We observe that all methods experience a noticeable performance degradation at the highest rank of 128 compared to ranks 32 and 64 on reasoning tasks. This trend is likely attributable to the substantially increased number of newly introduced parameters at higher ranks, which demands smaller learning rates and more careful optimization. Such sensitivity is further exacerbated by the limited training data available in several reasoning benchmarks, making higher-rank adaptation more prone to overfitting or unstable training.

# E. Full Results of Section 4.5.1

In this section, we report results on all reasoning datasets from the minibatch-size ablation study introduced in Section 4.5.1. Figures 6 and 7 summarize the performance of FILet under different minibatch sizes used to estimate the empirical second-moment statistics during initialization. Specifically, we evaluate minibatch sizes of 80, 160, 320, 480, and 640 for both

Llama2-7B and Llama3-8B, enabling a systematic examination of how the amount of data used for basis construction influences downstream adaptation performance. In addition, we include a reference setting in which the singular bases are computed using the full training dataset, denoted as "Full".

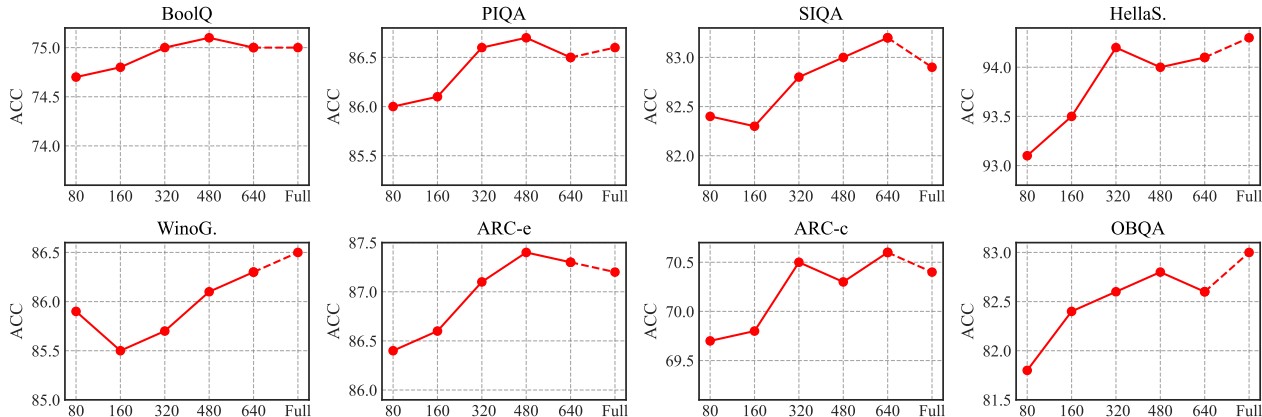

*Figure 6.* Full results of the ablation study on the Llama2-7B model.

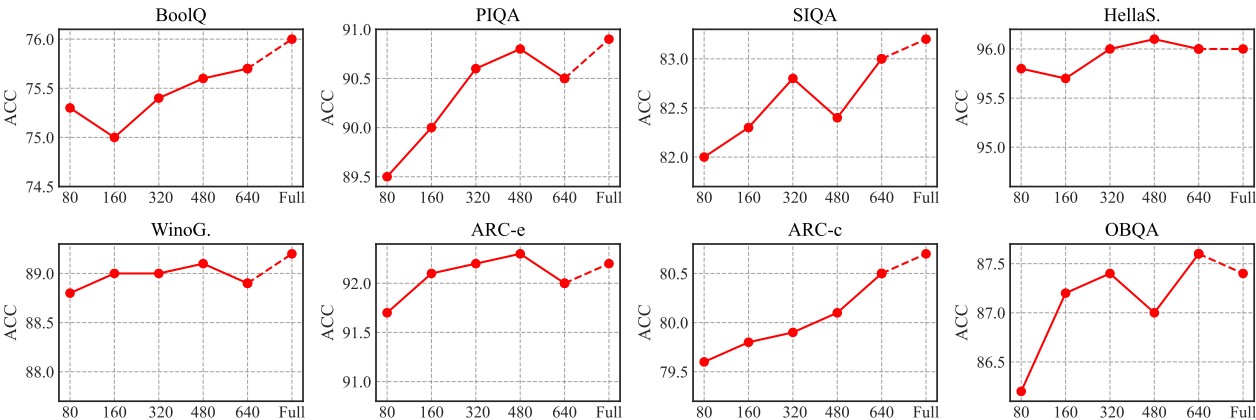

*Figure 7.* Full results of the ablation study on the Llama3-8B model.

The complete experimental results show that FILet exhibits robust and stable performance across a wide range of minibatch sizes. In general, increasing the minibatch size yields more accurate estimates of the empirical second-moment statistics, which in turn leads to improved downstream adaptation performance. Nevertheless, FILet remains competitive even when initialized with relatively small minibatches, achieving strong performance on most benchmarks. This behavior highlights the effectiveness and flexibility of FILet in practical settings, where computational resources or data availability for initialization may be constrained, and underscores its ability to balance estimation accuracy with computational efficiency.

## F. Full Results of Section 4.5.2

In this section, we report the complete results of the initialization strategy ablation study introduced in Section 4.5.2.

For comparison, we evaluate the following initialization strategies:

- **FILet-SVD-P**: Uses the directions with the maximal Fisher Energy obtained from the SVD of the pre-trained weights for both $U$ and $V$, with SVD-based scaling:

$$A = U_{[:,0:r]}\sqrt{\Sigma}_{[0:r,0:r]}, \quad B = \sqrt{\Sigma}_{[0:r,0:r]}V_{[:,0:r]}^{\top}. \tag{32}$$

- **FILet-P**: Uses the directions with the maximal Fisher Energy for both $\hat{U}$ and $\hat{V}$, with Fisher-based scaling. Specifically, we select the top-$r$ directions according to Fisher Energy:

$$\mathcal{I} = \text{TopK}(\Sigma_{\text{fisher}}, r), \quad \hat{U}_{\text{sel}} = \hat{U}[:, \mathcal{I}], \quad \hat{V}_{\text{sel}} = \hat{V}[:, \mathcal{I}], \quad \Sigma_{\text{sel}} = \Sigma_{\text{fisher}}[\mathcal{I}, \mathcal{I}], \tag{33}$$

and initialize

$$A = \hat{U}_{\text{sel}}\sqrt{\Sigma_{\text{sel}}}, \quad B = \sqrt{\Sigma_{\text{sel}}}\hat{V}_{\text{sel}}^{\top}. \tag{34}$$

- **FILet-SVD-R**: Uses randomly selected directions from the pre-trained weight matrices, scaled by their corresponding SVD singular values.

- **FILet-R**: Uses randomly selected directions from the pre-trained weight matrices, scaled according to their corresponding Fisher Energy values.

- **FILet-SVD-M**: Uses the directions with the minimal Fisher Energy from the SVD of the pre-trained weights for both $U$ and $V$, with SVD-based scaling:

$$A = U_{[:, h-r:h]}\sqrt{\Sigma}_{[h-r:h, h-r:h]}, \quad B = \sqrt{\Sigma}_{[h-r:h, h-r:h]}V_{[:, h-r:h]}^{\top}. \tag{35}$$

- **FILet-M**: The proposed FILet method, which selects the directions with the minimal Fisher Energy and applies Fisher-based scaling.

Here, $h = \min(m, n)$ denotes the total number of singular values of the pre-trained weight matrix.

Table 12 summarizes the performance of different FILet initialization variants on reasoning benchmarks, using both Llama2-7B and Llama3-8B as base models.

*Table 12.* Full experimental results for different initialization strategies on reasoning benchmarks using Llama2-7B and Llama3-8B as the base models. The best results for each model are highlighted in **bold**.

| Model | PEFT | **BoolQ** | **PIQA** | **SIQA** | **HellaS.** | **WinoG.** | **ARC-e** | **ARC-c** | **OBQA** | **Avg.** |
|---|---|---|---|---|---|---|---|---|---|---|
| | FILet-SVD-P | 73.4 | 84.3 | 81.5 | 93.7 | 83.6 | 85.0 | 64.3 | 81.2 | 80.9 |
| | FILet-P | 74.0 | 85.3 | 82.4 | 93.6 | 85.3 | 85.4 | 67.2 | 81.4 | 81.8 |
| Llama2-7b | FILet-SVD-R | 74.1 | 84.6 | 81.6 | 94.1 | 83.6 | 85.9 | 64.7 | 81.6 | 81.3 |
| | FILet-R | 74.8 | 85.9 | 82.2 | 94.0 | 85.3 | 86.6 | 68.8 | 82.2 | 82.5 |
| | FILet-SVD-M | 74.5 | 84.8 | 82.5 | 93.6 | 85.1 | 85.7 | 66.6 | 82.4 | 81.9 |
| | FILet-M | **75.0** | **86.6** | **82.8** | **94.2** | **85.7** | **87.1** | **70.5** | **82.6** | **83.1** |
| | FILet-SVD-P | 74.5 | 88.2 | 81.7 | 95.6 | 87.2 | 90.9 | 76.6 | 83.2 | 84.7 |
| | FILet-P | 75.0 | 89.0 | 82.2 | 95.9 | 88.9 | 91.3 | 78.8 | 85.6 | 85.8 |
| Llama3-8b | FILet-SVD-R | 74.8 | 88.0 | 81.8 | 95.2 | 87.8 | 90.7 | 77.2 | 83.4 | 84.8 |
| | FILet-R | 75.4 | 90.3 | **83.0** | 95.8 | 88.7 | 91.8 | 79.4 | 86.8 | 86.4 |
| | FILet-SVD-M | **75.6** | 89.4 | 82.3 | 95.9 | 88.8 | 91.6 | 78.0 | 85.0 | 85.8 |
| | FILet-M | 75.4 | **90.6** | 82.8 | **96.1** | **89.3** | **92.2** | **79.9** | **87.4** | **86.7** |

Our proposed FILet method (FILet-M) selects the least-dominant Fisher-aligned directions and applies Fisher-based scaling that is explicitly matched to the local Fisher Energy landscape, while maintaining minimal magnitude to preserve the structure of the pre-trained weights. The superior performance of FILet-M can be attributed to two key factors: (1) Fisher-aligned bases more accurately identify and align with directions that are most beneficial for reducing the loss, and (2) minimal Fisher-energy scaling avoids unnecessary distortion of the pre-trained parameters during initialization, thereby mitigating information loss and enabling more stable and effective fine-tuning.

Table 13 summarizes the performance of different FILet initialization variants on image classification benchmarks, using ViT-B/32 as the base model.

The results on image classification benchmarks exhibit trends consistent with those observed in reasoning tasks. In particular, FILet-M achieves the highest average performance, outperforming all other initialization variants.

*Table 13.* Full experimental results for different initialization strategies on image classification benchmarks using ViT-B/32 as the base model. The best results are highlighted in **bold**.

| PEFT | Cars | DTD | EuroSAT | GTSRB | RESISC45 | SUN397 | SVHN | Avg. |
|------|------|------|---------|-------|----------|--------|------|------|
| FILet-SVD-P | 76.6 | 68.5 | 88.7 | 51.9 | 76.2 | 75.8 | 39.9 | 68.3 |
| FILet-P | 76.9 | 68.4 | 88.8 | 51.8 | 76.1 | 75.9 | 40.8 | 68.4 |
| FILet-SVD-R | 76.6 | 68.5 | **89.2** | **52.5** | 76.4 | 76.0 | 40.9 | 68.6 |
| FILet-R | 79.0 | 69.0 | 88.9 | 52.0 | 76.2 | 76.0 | 41.2 | 68.9 |
| FILet-SVD-M | 76.1 | **69.8** | **89.2** | 51.2 | 76.4 | **76.1** | 39.6 | 68.3 |
| FILet | **79.2** | 69.5 | 89.5 | **52.5** | **76.5** | **76.1** | **41.6** | **69.3** |

# G. Direction Selection Analysis

In this section, we present additional visualizations analyzing the overlap of selected adaptation directions across different tasks. Figures 8 and 9 show the direction overlap matrices for a range of tasks using Llama2-7B and Llama3-8B as base models, respectively. Each entry in the matrix measures the degree of alignment between the sets of adaptation directions selected for a pair of tasks.

Specifically, for each task and each adapted module, we compute the $r$ directions selected by the initialization method, and then quantify the overlap between the corresponding direction sets for every pair of tasks. The reported values represent the average overlap percentage across all adapted modules, providing a holistic view of how task-specific adaptation subspaces align or diverge across different downstream objectives.

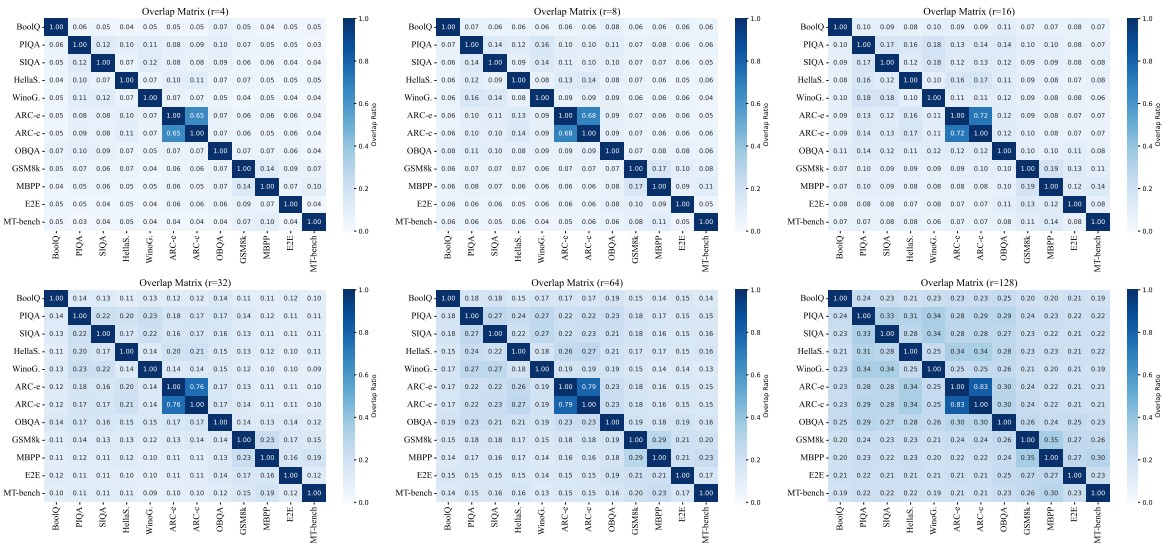

*Figure 8.* Direction overlap matrices different tasks using Llama2-7B as the base model.

From these visualizations, we observe that "ARC-e" and "ARC-c" exhibit a notably high degree of direction overlap, which is expected since they are essentially two subsets of the same benchmark. Beyond this pair, most task combinations display relatively low overlap in their selected adaptation directions, indicating that FILet captures task-specific characteristics in the adaptation subspaces.

This behavior can be understood through the interaction between the data distribution and the model architecture. The Fisher Energy is inherently dependent on both factors: while different tasks induce substantially different data distributions and therefore distinct Fisher Energy landscapes, the underlying model architecture remains fixed across tasks. As a result, certain structural properties of the Fisher Energy are shared, causing some directions to be consistently identified as important across tasks. At the same time, task-specific data distributions emphasize different regions of the parameter space, leading to largely distinct sets of selected directions and explaining the overall low overlap observed among most task pairs.

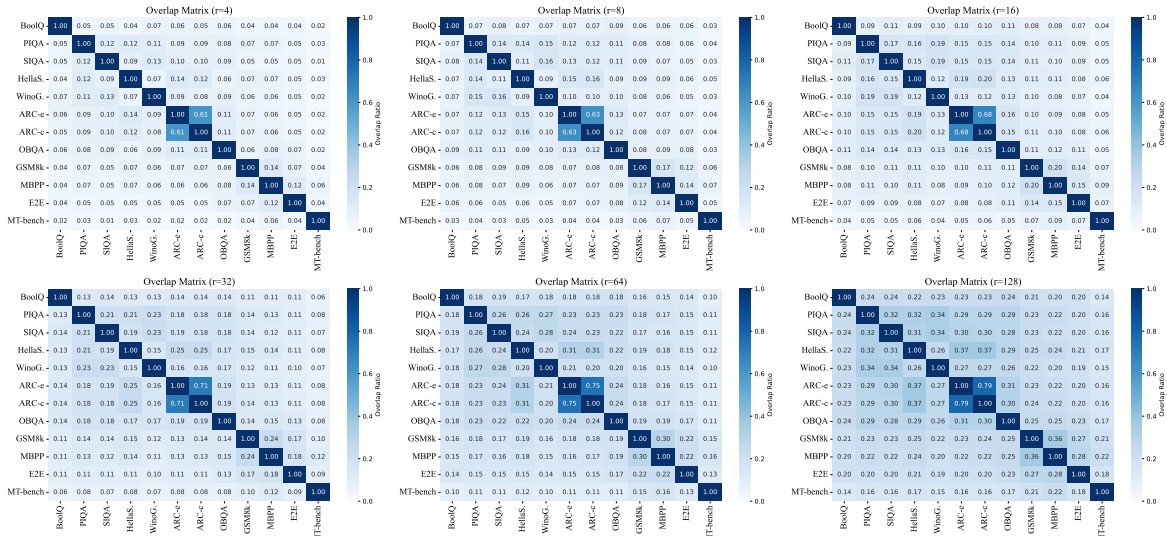

*Figure 9.* Direction overlap matrices different tasks using Llama3-8B as the base model.

## H. Limitations

Compared to SVD-based initialization methods, FILet incurs additional memory overhead during the initialization phase to compute and store empirical second-moment statistics. While this overhead is not significant in most scenarios, it may become a practical challenge when adapting extremely large models or deploying on hardware with stringent memory constraints. An important direction for future work is to develop more memory-efficient approximations, such as low-precision, blockwise, or streaming estimators of the second-moment statistics, to further reduce the initialization footprint and broaden the applicability of FILet in resource-limited settings.

