# OpenReview forum: "Learning in the Fisher Subspace: A Guided Initialization for LoRA Fine-Tuning"
_ICML.cc/2026/Conference — ICML 2026 regular_

### Official Review · Reviewer_NaTa · 2026-03-08

**Soundness:** 3
**Presentation:** 3
**Significance:** 2
**Originality:** 3
**Overall Recommendation:** 4
**Confidence:** 3

**Summary:**

The authors address a critical limitation in LoRA: the reliance on task-agnostic, weight-only initialization strategies. They argue that the effectiveness of adaptation critically depends on which subspace is chosen at initialization; however, task-irrelevant directions can cause sub-optimal downstream performance.

To mitigate this, the paper proposes a data-aware initialization framework. By leveraging Fisher information to capture the curvature induced by the target data distribution, the method identifies how parameter perturbations influence model predictions, thereby providing a task-dependent criterion for selecting LoRA directions for the downstream task.

**Compliance With Llm Reviewing Policy:**

Affirmed.

**Final Justification:**

My concerns have been partially addressed; however, gaps remain in fully resolving them, particularly regarding generalizability and scalability (especially across different fine-tuning setups).

**Key Questions For Authors:**

1. Could the authors evaluate the downstream performance on more recent or smaller architectures (e.g., <3B models) to demonstrate broader applicability.

2. Forgetting of prior knowledge: It would be interesting to analyze how the method affects forgetting. For example, in Table 1, the authors could additionally report performance on the native benchmark tasks used in the original papers to assess potential forgetting.

3. I am curious how the method behaves under different domain gaps. For example, in Table 2, Using ViT-B/32, could the authors evaluate fine-tuning on a highly specialized domain (e.g., medical images), which would involve a much larger domain shift?

4. Does this method extend to multimodal LLMs as well?

5. Please analyze the impact of minibatch distribution bias. If the minibatch data is biased, does the method perform poorly? If so, how can this issue be mitigated or prevented in practice?

6. Comparison with recent work: Please include a comparison with LoRA-GA [1].

[1] LoRA-GA: Low-Rank Adaptation with Gradient Approximation (NeurIPS 2024)

**Limitations:**

Same as the weaknesses.

**Strengths And Weaknesses:**

1. Strengths

- Clear and convincing explanation of why a data-driven approach is necessary.

- The proposed algorithm is convincing.

- Clear flow from motivation to design with the evidence (Figure 1)


2. Weaknesses

- Overall evaluation is shallow: limited generality and limited scalability.

- Effect on forgetting of prior knowledge is not sufficiently analyzed.

- Comparison with recent work (e.g., LoRA-GA) is missing or insufficient.

---

> ### Author Rebuttal · Authors · 2026-03-29
>
> Thank you for your valuable suggestions. Below are our response towards these concern.
>
> 1. Weakness 1, Question 1
>
> We agree that evaluating across diverse model scales would further demonstrate the generality of our method.
>
> Our current evaluation already spans a broad set of representative modern architectures, including Llama2-7B, Llama3-8B, Gemma-7B, Qwen2.5-7B, and ViT. These models differ in training recipes, tokenization, and architectural design, and are widely used, providing strong evidence of the architecture-agnostic nature of our work. Importantly, our method is inherently scale-independent, as it relies on layer-wise statistics rather than assumptions about model size. We focused on larger, widely used models since they offer a more challenging and practically relevant testbed; thus, improvements on 7B–8B models provide strong validation of effectiveness.
>
> 2. Weakness 2, Question 2
>
> We agree that analyzing forgetting is important when evaluating fine-tuning methods. However, our work focuses on initialization for PEFT rather than continual learning or full-model adaptation. In LoRA-style PEFT, the pretrained backbone is frozen and only a small set of low-rank parameters is trained, which prior work has shown to inherently mitigate catastrophic forgetting compared to full fine-tuning.
>
> Methodologically, our Fisher-guided initialization only affects the initial subspace and scaling of low-rank updates, without modifying backbone weights or increasing adaptation capacity. Therefore, it is unlikely to introduce additional forgetting beyond standard LoRA. In fact, by aligning updates with Fisher-informed sensitivities, it may better preserve pretrained knowledge by avoiding highly sensitive directions.
>
> 3. Weakness 3, Question 6
>
> LoRA-GA is an initialization method that aligns low-rank updates with full fine-tuning gradients via direction selection and scaling. We reproduce this approach, and report results on Llama2-7B below:
>
> ||BoolQ|PIQA|SIQA|HellaS.|WinoG.|ARC-e|ARC-c|OBQA|Avg.|
> |-|-|-|-|-|-|-|-|-|-|
> |LoRA-GA|74.4|85.1|82.5|93.1|84.1|84.2|65.9 |82.0|81.4|
> |FILet|75.0|86.6|82.8|94.2|85.7|87.1|70.5 |82.6|83.1|
>
> FILet outperforms LoRA-GA on reasoning benchmarks. Notably, the two methods target different aspects of LoRA optimization: LoRA-GA focuses on improving convergence via better parameter initialization, while FILet identifies task-insensitive subspaces and preserves pretrained knowledge in residual weights, enabling more effective adaptation and improved convergence.
>
> 4. Question 3
>
> We agree that robustness under domain shift is important to evaluate.
>
> Our experiments already involve substantial domain shifts. In particular, math reasoning and code generation differ significantly from typical pretraining distributions, requiring structured reasoning, symbolic manipulation, and precise syntax. As shown in Table 3, FILet consistently improves over baselines across multiple models in these settings, indicating strong performance under non-trivial domain gaps. Methodologically, this robustness stems from FILet’s design: it identifies subspaces with low task-relevant Fisher energy and allocates adaptation capacity to these directions, while preserving pretrained knowledge in the residual weights. Under larger shifts, this enables efficient capacity reallocation without disrupting critical features.
>
> We agree that highly specialized domains are an interesting direction. However, evaluating them would require additional domain-specific datasets and pipelines beyond our current scope, and we leave this for future work.
>
> 5. Question 4
>
> While we do not evaluate multimodal LLMs, our method is not limited to unimodal settings. It operates at the level of individual linear layers, using activation statistics and output gradients to construct Fisher-guided low-rank subspaces—components that also exist in multimodal architectures. Thus, it can be applied without architectural changes.
>
> That said, multimodal models introduce additional complexities (e.g., modality alignment and heterogeneous features), and thorough evaluation is beyond our scope. We leave this as future work.
>
> 6. Question 5
>
> We agree that minibatch quality can affect the estimation of our statistics.
>
> Our method relies on estimating activation and gradient factors ($S_X$, $S_Y$) from sampled minibatches, so distribution bias manifests as estimation bias or variance. We study this via an ablation on minibatch size (Figure 5), varying samples from 80 to the full dataset. Results show consistent gains with larger batches, indicating that better estimates improve subspace quality. Importantly, performance degrades gradually with smaller batches, not catastrophically, suggesting robustness to imperfect sampling.
>
> In practice, this issue can be alleviated by:
> (1) using larger or more diverse minibatches,
> (2) optionally using a small held-out subset that better reflects the target task distribution.

---

> > ### Author Rebuttal · Reviewer_NaTa · 2026-04-02
> >
> > I thank the authors for their responses. However, there are still gaps in fully addressing my concerns, particularly regarding generalizability and scalability (especially handling different fine-tuning setups). Therefore, I will maintain my current rating of 4.

---

### Official Review · Reviewer_psV5 · 2026-03-09

**Soundness:** 2
**Presentation:** 3
**Significance:** 3
**Originality:** 3
**Overall Recommendation:** 3
**Confidence:** 4

**Summary:**

This paper proposes a LoRA initialisation method called FILet. The core idea is to use data-driven Fisher information to replace previous methods that rely solely on weight matrix for direction selection. Starting from Equation 2, which aims to minimise the expected increase in loss, the authors use approximations to derive a formula that minimises Fisher Energy (Equation 7). The paper then introduces the FILet algorithm, which uses the K-FAC approximation for Fisher information and selects the directions with the lowest Fisher Energy as the initial LoRA directions.

**Compliance With Llm Reviewing Policy:**

Affirmed.

**Final Justification:**

I appreciate the authors' acknowledgement that the statement regarding first-order terms cancelling out under symmetric perturbation is confusing.However, this reveals a stark contradiction. In the manuscript, the authors explicitly state after Equation 2: ''Our objective is to select directions (u, v) that minimize the expected increase in loss, i.e., those that most effectively reduce $\Delta \mathcal{L}$ in expectation''. Conversely, in the reply rebuttal, the authors concede: ''Our intention is not to strictly minimize Eq.~(2), but to use the Taylor expansion as a motivation''. This contradiction fundamentally compromises the theoretical framing in Section 3.1, as the motivational logic in this section requires a major rewrite to be logically coherent. While I acknowledge the authors' attempt to patch this theoretical gap in their rebuttal by introducing the new $\Delta \mathcal{L}_{\text{sym}}(Z)$ formulation, this represents a fundamental shift in the paper's core motivational logic rather than a minor clarification. Such a major theoretical reconstruction should undergo a full rigorous peer-review cycle rather than being patched during the rebuttal phase. Therefore, I have decided to maintain my current score.

**Key Questions For Authors:**

1. Could you detail why the intuitive meaning remains consistent after approximating Equation 2 to arrive at Equation 7?
2. Could you elaborate on why it is justifiable to base your approximation on a 'symmetric perturbation' when practical fine-tuning relies on directional gradient descent?
3. Could you explain the specific conditions under which approximating the Hessian with the Fisher information matrix holds true (for Equation 6), and discuss whether those conditions are actually met during the initialisation phase of fine-tuning?
4. What is the core limitation of LoRA-Dash, and how does your paper build upon, differ from, and improve upon it?
5. Could you provide more detailed statistical information (such as mean and standard deviation) or conduct significance testing to prove that FILet's performance gains are not coincidental?
6. Could you provide quantitative charts comparing the peak VRAM usage of FILet against the baseline methods?

**Limitations:**

Yes.

**Strengths And Weaknesses:**

Strengths
1. LoRA initialisation is a highly relevant and important research problem.
2. Leveraging data-aware sensitivity for LoRA initialisation is a very interesting and promising direction.

Weaknesses
1. Intuitive inconsistency between Equation 2 and Equation 7
The paper starts by aiming to minimise the expected increase in loss (Equation 2). Through several approximations, it concludes that one should minimise Fisher Energy (Equation 7). However, if a direction has extremely low Fisher Energy, it implies the loss landscape is exceptionally flat in that direction, with a very weak gradient signal. The fundamental goal of fine-tuning is to rapidly decrease the loss, not merely to safely avoid increasing it.
2. Questionable approximation in Equation 4:
The paper justifies ignoring the first-order gradient by stating, "If we consider a symmetric perturbation along direction Z, the first-order terms cancel out". In actual training, many optimisers move strictly along the first-order negative gradient to minimise loss; they do not blindly jump back and forth in both positive and negative directions. Therefore, for the downstream task the model is genuinely trying to learn, the first-order term is absolutely dominant. Using a symmetric perturbation to justify this approximation seems problematic.
3. Unjustified approximation in Equation 6:
Approximating the Hessian matrix with the empirical Fisher information matrix only holds true under specific theoretical conditions. The paper does not explain or justify why this approximation is valid in the context of early-stage fine-tuning.
4. Comparative discussion of closely related reference
The experimental section introduces LoRA-Dash as a "representative data-driven LoRA enhancement baseline". However, this closely related reference is completely missing from the Related Work section, lacking a proper comparative discussion.
5. Lack of statistical significance: In Tables 1 and 2, the paper reports the median over five runs. However, it does not provide the variance or perform any significance testing, making it difficult to assess the reliability of the improvements.
6. Omission of VRAM comparisons: Calculating the gradient covariance theoretically introduces a significant peak VRAM overhead. The authors briefly mention that FILet incurs "additional memory overhead" in a single sentence in Appendix H (Limitations), but there are no quantitative charts comparing VRAM usage across methods.

---

> ### Author Rebuttal · Authors · 2026-03-29
>
> We thank the reviewer for the careful reading and constructive feedback. Below are our responces:
>
> 1. Response to Weakness 1 and 2
>
> A rapid decrease in loss is indeed a goal of fine-tuning. However, in LoRA initialization, we are not explicitly selecting directions that the parameters will follow during optimization. From the perspective of initialization, we decompose the pretrained weight into two parts: one that will be updated and one that remains preserved in the residual weights. Therefore, the goal is to identify a subspace that contains the least task-relevant information. In other words, modifying this subspace should incur minimal risk of increasing the loss, which is the key idea behind our approach.
>
> By applying a Taylor expansion to eq. (2), we decompose the loss change into two main components: the first-order term, which captures the gradient direction, and the second-order term, which reflects sensitivity. Since the gradient direction is naturally followed during fine-tuning, we instead focus on identifying directions that minimize the second-order term. Our goal is to identify low-risk directions, and let the fine-tuning naturally follows the gradient direction during optimization, which is the key reason of why we cancel the first-order term
>
> 2. Response to Weakness 3
>
> We agree that the empirical Fisher is not identical to the Hessian, and will clarify that Eq. (6) serves as a local second-order proxy rather than an exact equality. In our work, this approximation is used only for initialization—specifically, to rank directions under small perturbations based on a second-order Taylor expansion.
>
> To reduce computational cost, we leverage efficient factorizations of the Fisher Information Matrix and use only diagonal components in a projected subspace, avoiding explicit construction of the full curvature matrix. This mitigates discrepancies with the Hessian while remaining efficient.
>
> More broadly, FILet is an approximation pipeline: it combines empirical Fisher estimates with K-FAC factorization to obtain a tractable sensitivity signal. Thus, it does not rely on strict Hessian–Fisher equivalence, but only on a practical curvature-informed criterion for initialization, which is supported by our empirical results.
>
> 3. Response to Weakness 4
>
> LoRA-Dash proposes a training-time enhancement for LoRA: it performs a short warm-up fine-tuning, then applies SVD to the LoRA weights to extract task-specific directions (TSD) for reparameterization, aiming to accelerate convergence and improve performance. However, it has several limitations. It depends on early training dynamics, which can be noisy or unstable, especially in low-data settings. It also relies on SVD of intermediate weights without explicitly incorporating input–output statistics or curvature, and thus does not directly address initialization quality.
>
> In contrast, FILet is a principled initialization method rather than a training-time enhancement. It leverages Fisher-informed statistics to characterize model structure, viewing initialization as a denoising step on pretrained weights. By selecting directions with low Fisher energy, FILet preserves pretrained knowledge while enabling effective adaptation, which is fundamentally different from LoRA-Dash.
>
> 4. Response to Weakness 5
>
> We reorganize our experimental results and report the standard deviations for FILet and the reproduced baselines. The results are shown below:
>
> ||BoolQ|PIQA|SIQA|HellaS.|WinoG.|ARC-e|ARC-c|OBQA|Avg.|
> |-|-|-|-|-|-|-|-|-|-|
> |KaSA(Llama2-7b)|74.3(0.55)|85.2(0.66)|81.9(0.32)|93.4(0.23)|85.3(0.59)|86.3(0.62)|67.4(0.87)|80.2(0.77)|81.7(0.22)|
> |LoRA-Dash(Llama2-7b)|73.9(0.58)|85.0(0.79)|82.0(0.38)|94.2(0.19)|84.3(0.90)|85.7(0.49)|68.6(0.92)|79.4(0.79)|81.7(0.38)|
> |FILet(Llama2-7b)|75.0(0.40)|86.6(0.50)|82.8(0.50)|94.2(0.22)|85.7(0.53)|87.1(0.58)|70.5(0.81)|82.6(0.62)|83.1(0.26)|
> |KaSA(Llama3-8b)|74.3(0.71)|87.8(0.70)|82.2(0.55)|95.2(0.38)|87.5(0.54)|91.6(0.57)|80.3(0.63)|87.0(0.62)|85.7(0.23)|
> |LoRA-Dash(Llama3-8b)|74.7(0.67)|87.4(0.53)|82.0(0.42)|95.8(0.30)|87.2(0.55)|92.0(0.43)|80.0(0.66)|86.0(1.06)|85.6(0.29)|
> |FILet(Llama3-8b)|75.4(0.55)|90.6(0.57)|82.8(0.52)|96.1(0.28)|89.0(0.45)|92.2(0.58)|79.9(0.39)|87.4(0.61)|86.7(0.13)|
>
>
> The values in the subscripts denote standard deviation. From these results, FILet demonstrates stable performance and consistently outperforms the baselines.
>
> 5. Response to Weakness 6
>
> In our experimental setting, the memory requirements (GBs) of LoRA and FILet initialization and training are shown as below.
>
> ||BoolQ|PIQA|SIQA|HellaS.|WinoG.|ARC-e|ARC-c|OBQA|
> |-|-|-|-|-|-|-|-|-|
> |LoRA(Llama2-7B)|24.9|52.1|35.2|44.1|26.9|33.8|33.5|29.8|
> |FILet(Llama2-7B)|32.9|65.3|43.0|52.2|37.0|45.5|45.2|38.9|
> |LoRA(Llama3-8B)|26.5|50.3|33.4|52.2|28.3|34.7|35.2|30.3|
> |FILet(Llama3-8B)|36.1|63.0|42.5|62.4|39.3|47.3|47.8|41.5|
>
> FILet uses more memory during initialization to cache intermediate values but consistently delivers much better performance.

---

> > ### Author Rebuttal · Reviewer_psV5 · 2026-04-02
> >
> > Thank you for your detailed responses to my initial concerns.
> >
> > Following your replies to Weaknesses 1 and 2, I have two further questions. In your rebuttal, you state: "we decompose the loss change into two main components: the first-order term, which captures the gradient direction, and the second-order term, which reflects sensitivity. **Since the gradient direction is naturally followed during fine-tuning, we instead focus on identifying directions that minimize the second-order term.**"
> >
> > Question 1: In the manuscript, you state: "If we consider a symmetric perturbation along direction Z, the first-order terms cancel out." Could you please explain the contradiction here? The paper claims the first-order terms mathematically cancel out, whereas your rebuttal suggests you simply chose to ignore them as a heuristic to focus on the second-order term.
> >
> > Question 2: If focusing on the second-order term is indeed a design choice, the mathematical derivation in Section 3.1 lacks rigour. Why is Equation 2 (which aims to minimise the expected change in downstream loss) used as the formal starting point, only to neglect the first-order term in order to reach Equation 7?

---

> > > ### Author Response · Authors · 2026-04-03
> > >
> > > We thank the reviewer for pointing out this inconsistency. We agree that this is a valid concern and that the statements in Section 3.1 may be misleading.
> > >
> > > 1. Question 1:
> > >
> > > We agree that the statement in the manuscript that ``the first-order terms cancel out under symmetric perturbation'' may be confusing. In practice, fine-tuning follows the negative gradient direction, and the first-order term is generally dominant for small steps. Our original intention was to use perturbation only as a tool to isolate curvature (second-order) effects when evaluating the sensitivity of a direction, rather than to describe the actual optimization dynamics. We will revise the manuscript and clarify that the first-order term does not vanish, but we choose to focus on the second term in the camera-ready version.
> > >
> > > 2. Question 2:
> > > Our primary goal in Section 3.1 and 3.2 is to justify:
> > > (1) Why do we aim to select directions with low (rather than high) Fisher energy, and
> > > (2) why the Fisher energy is defined as $\mathcal{E}(Z) = \mathrm{vec}(Z)^\top S_W\,\mathrm{vec}(Z)$
> > >
> > > Our intention is not to strictly minimize Eq.~(2), but to use the Taylor expansion as a motivation to separate two roles in fine-tuning:
> > > The first-order term captures the descent direction, while the second-order term reflects the sensitivity (curvature) of the loss landscape.
> > >
> > > In the fine-tuning stage, parameter updates naturally follow the negative gradient direction. Therefore, in the initialization stage, our focus is to identify a subspace with low sensitivity, such that updates within this subspace are less likely to incur large loss increases. This motivates selecting directions with low curvature (low Fisher energy), which can lead to more effective optimization trajectories, as supported by the experiments in Section 3.2.
> > >
> > > The perturbation along direction $Z$ considered here is primarily used in the initialization stage. At this stage, our objective is to exploit structural information of the loss landscape that cannot be captured by standard gradient-based optimization. This motivates us to focus on the second-order term. Inspired by prior work~[1], we therefore adopt a second-order approximation to characterize directional sensitivity and guide our method.
> > >
> > > **Revised interpretation.**
> > >
> > > To clarify this point, we will revise the derivation to explicitly introduce the symmetric perturbation.
> > >
> > > Applying a second-order Taylor expansion of the loss around $W_0$ yields Eq. (3). To isolate the influence of the gradient direction and better measure sensitivity along direction $Z$, we consider a symmetric perturbation, leading to:
> > > $$
> > > \Delta \mathcal{L}_{\mathrm{sym}}(Z) := \frac{1}{2}[ \mathcal{L}(W_0 + \gamma Z) + \mathcal{L}(W_0 - \gamma Z) ] - \mathcal{L}(W_0).
> > > $$
> > >
> > > Under this symmetric construction, the first-order terms cancel out by design, yielding
> > >
> > > $$
> > > \Delta \mathcal{L}_{\mathrm{sym}}(Z) = \frac{\gamma^2}{2} < Z, (\nabla_{W_0}^2 \mathcal{L}) Z > + o(\gamma^2).
> > > $$
> > >
> > > This formulation isolates the second-order curvature of the loss landscape, yielding a direction-wise sensitivity measure independent of the downstream gradient. We hope this revision addresses your concern.
> > >
> > > We will revise the manuscript to
> > >
> > > (1) rephrase the statement about first-order cancellation,
> > >
> > > (2) clarify that focusing on the second-order term is a design choice motivated by subspace-constrained optimization rather than a direct consequence of Eq.~(2),
> > >
> > > (3) improve the overall presentation to clearly distinguish between the role of initialization and the role of optimization dynamics.
> > >
> > > [1] Pascanu, R., & Bengio, Y. (2013). Revisiting natural gradient for deep networks. arXiv preprint arXiv:1301.3584.

---

### Official Review · Reviewer_Maxm · 2026-03-10

**Soundness:** 3
**Presentation:** 3
**Significance:** 2
**Originality:** 2
**Overall Recommendation:** 5
**Confidence:** 4

**Summary:**

This paper proposes a data-dependent LoRA initialization scheme. The method computes an SVD of the target weight matrices and estimates relevant directions under the downstream data distribution using Fisher-related statistics. Those directions are then used to initialize the LoRA matrices. The authors report improvements over existing baselines across multiple language benchmarks (reasoning and generation) and ViT image classification.

**Compliance With Llm Reviewing Policy:**

Affirmed.

**Final Justification:**

The remaining concerns raised in my review were addressed by the authors. I therefore recommend the papers acceptance.

**Key Questions For Authors:**

See Weaknesses

**Limitations:**

yes

**Strengths And Weaknesses:**

Strengths:
- Clear motivation that weight-only SVD-based initializations can miss task-relevant directions. Data aware LoRA init is an interesting concept. Especially using Kronecker-factored (K-FAC-style) statistics to guide direction selection/initialization is novel to my knowledge of the related literature.
- Broad empirical results on reasoning, generation, and image classification tasks, with FILet reported as consistently outperforming baselines.
- Practical efficiency: Chapter 4.4 argues SVD baselines are dominated by FP32 SVD, while FILet reduces to matrix multiplications and avoids full SVD
- Minibatch-size ablation supports the method’s robustness

Weaknesses:
- Baseline comparability: Table 1 explicitly states results marked with a cross are taken from original papers. Even if some baselines are rerun, the paper would benefit from a clean vanilla LoRA comparison under the exact same training and eval setting. It is understandable that it might be too resource intensive to rerun baselines but at least vanilla lora would be good to rerun.
- Initialization speed: It would be interesting to have a wallclock comparison of the initialization overhead / improvements over baselines.
- Invariance to data ordering: Many results are reported as “median over five runs,” but not variance. Adding standard deviation would help assess sensitivity to minibatch sampling/order used during initialization.
- Related work gap: Section 2 discusses SVD-based approaches largely through SVD on pretrained weights. It would be important to cite Explained Variance Adaptation [1], which uses (incremental) SVD on minibatches of activation vectors for LoRA initialization.

I am willing to raise my score if my concerns are addressed.

References:

[1] Paischer, F., Hauzenberger, L., Schmied, T., Alkin, B., Deisenroth, M. P., & Hochreiter, S. (2025). Parameter efficient fine-tuning via explained variance adaptation. The Thirty-ninth Annual Conference on Neural Information Processing Systems.

---

> ### Author Rebuttal · Authors · 2026-03-29
>
> We thank the reviewer for the constructive suggestions. Below we address each concern in turn.
>
> 1. Response to Weakness 1
>
> In response to the reviewer’s concern, we reran vanilla LoRA under the exact same training and evaluation pipeline as FILet on both Llama2-7B and Llama3-8B.
>
> For Llama2-7b:
> ||BoolQ|PIQA|SIQA|HellaS.|WinoG.|ARC-e|ARC-c|OBQA|Avg.|
> |-|-|-|-|-|-|-|-|-|-|
> |LoRA*|69.8 |79.9|79.5|83.6   |82.6  |79.8 |64.7 |81.0 |77.6|
> |LoRA |74.2(0.49)|84.6(0.51)|81.9(0.53)|92.0(0.68)|83.3(0.47)|84.6(0.50)|66.2(1.16)|81.0(0.95)|81.0(0.29)|
> |FILet|75.0(0.40)|86.6(0.50)|82.8(0.50)|94.2(0.22)|85.7(0.53)|87.1(0.58)|70.5(0.81)|82.6(0.62)|83.1(0.26)|
>
> For Llama3-8b:
> ||BoolQ|PIQA|SIQA|HellaS.|WinoG.|ARC-e|ARC-c|OBQA|Avg.|
> |-|-|-|-|-|-|-|-|-|-|
> |LoRA*|70.8 |85.2|79.9|91.7|84.3|84.2|71.2|79.0|80.8|
> |LoRA|75.1(0.56)|88.4(0.56)|82.5(0.39)|94.6(0.49)|87.1(0.62)|90.3(0.66)|77.6(0.89)|84.4(1.03)|85.0(0.16)|
> |FILet|75.4(0.55)|90.6(0.57)|82.8(0.52)|96.1(0.28)|89.0(0.45)|92.2(0.58)|79.9(0.39)|87.4(0.61)|86.7(0.13)|
>
> LoRA* denotes the experimental results reported in the DoRA paper. We additionally report the standard deviations of FILet and LoRA as subscripts here. From this reproduced result, we observe that standard LoRA outperforms the reported LoRA* results. The rerun confirms that, even under the same setting, FILet consistently outperforms vanilla LoRA.
>
> 2. Response to Weakness 2
>
> We thank the reviewer for this suggestion. We actually include a wall-clock comparison in Section 4.4 / Figure 4, where we report the extra initialization time in seconds.
> The actual value (in seconds) is shown in the table below:
>
> | |1B|3B|8B|
> |-|--|--|--|
> |filet80|2.81|5.70|15.33|
> |filet320|9.94|28.38|53.04|
> |filet640|19.35|40.11|103.24|
> |kasa|21.25|84.20|217.32|
> |dash|23.22|88.66|225.45|
>
>
> These results show that FILet has substantially lower initialization overhead than KaSA and LoRA-Dash, and the gap becomes larger as model size increases. All timings were measured on a single NVIDIA A100 GPU, rank 32, and BF16 precision.
>
>
> 3. Response to Weakness 3
>
> We reorganize our experimental results and report the standard deviations for FILet and the reproduced baselines. The results are shown below:
>
> For Llama2-7b:
> ||BoolQ|PIQA|SIQA|HellaS.|WinoG.|ARC-e|ARC-c|OBQA|Avg.|
> |-|-|-|-|-|-|-|-|-|-|
> |KaSA|74.3(0.55)|85.2(0.66)|81.9(0.32)|93.4(0.23)|85.3(0.59)|86.3(0.62)|67.4(0.87)|80.2(0.77)|81.7(0.22)|
> |LoRA-Dash|73.9(0.58)|85.0(0.79)|82.0(0.38)|94.2(0.19)|84.3(0.90)|85.7(0.49)|68.6(0.92)|79.4(0.79)|81.7(0.38)|
> |FILet|75.0(0.40)|86.6(0.50)|82.8(0.50)|94.2(0.22)|85.7(0.53)|87.1(0.58)|70.5(0.81)|82.6(0.62)|83.1(0.26)|
>
> For Llama3-8b:
>
> ||BoolQ|PIQA|SIQA|HellaS.|WinoG.|ARC-e|ARC-c|OBQA|Avg.|
> |-|-|-|-|-|-|-|-|-|-|
> |KaSA|74.3(0.71)|87.8(0.70)|82.2(0.55)|95.2(0.38)|87.5(0.54)|91.6(0.57)|80.3(0.63)|87.0(0.62)|85.7(0.23)|
> |LoRA-Dash|74.7(0.67)|87.4(0.53)|82.0(0.42)|95.8(0.30)|87.2(0.55)|92.0(0.43)|80.0(0.66)|86.0(1.06)|85.6(0.29)|
> |FILet|75.4(0.55)|90.6(0.57)|82.8(0.52)|96.1(0.28)|89.0(0.45)|92.2(0.58)|79.9(0.39)|87.4(0.61)|86.7(0.13)|
>
> These results show that FILet exhibits stable performance, consistently outperforms the baselines, and is not overly sensitive to the minibatch sampling order during initialization.
>
> 4. Response to Weakness 4
>
> Thank you for pointing this out. EVA is indeed an important related method and should be discussed in Section 2. We will add it in the revision.
> EVA and FILet are both data-aware LoRA methods, but they differ in both goal and mechanism. EVA performs SVD on minibatch activation vectors and emphasizes explained variance and rank reallocation, whereas FILet uses Fisher-informed second-moment statistics to select initialization directions with low Fisher energy. Therefore, EVA is closely related and worth citing, and it is not redundant with our method.

---

> > ### Author Rebuttal · Reviewer_Maxm · 2026-04-02
> >
> > I thank the authors for their detailed rebuttal, which addresses my concerns clearly. I will raise my score to 5.

---

### Official Review · Reviewer_tkuh · 2026-03-12

**Soundness:** 3
**Presentation:** 2
**Significance:** 3
**Originality:** 3
**Overall Recommendation:** 4
**Confidence:** 4

**Summary:**

The paper provides an empirical analysis of initialization strategies for LoRA adapters and proposes a Fisher-informed approach to select and scale adaptation directions derived from the pre-trained weight matrix of a particular component. The method frames LoRA initialization as identifying parameter-space directions that are impactful under the downstream data distribution, using Fisher information (K-FAC approximation) to capture data-induced curvature of the loss landscape.

**Compliance With Llm Reviewing Policy:**

Affirmed.

**Final Justification:**

The rebuttal addressed my main concerns. Standard deviations are now reported and confirm that the improvements are consistent and meaningful. Memory overhead is quantified, making the trade-off transparent.
My residual concern is with the theoretical exposition around the first-order term in Eq. (3), which the rebuttal clarifies is deliberately ignored rather than formally canceled, a distinction that should be stated more carefully in the revision to avoid misleading readers. The presentation issues noted in my review (notation, repeated sentences, figure clarity) are minor and can be resolved in the camera-ready version.

**Key Questions For Authors:**

1. Can you provide a formal definition of the "symmetric perturbation along direction $Z$" described in L141--L144? Under what assumptions do the first-order terms cancel?
2. How are the specific singular values in Eq. (8) selected? What does the index $i$ represent?
3. How does Eq. (8) translate to initialization? Does it follow Eq. (1)?
4. Do you envision extending this approach beyond initialization, e.g., by adapting the LoRA subspace during fine-tuning based on Fisher information? Would such an approach be feasible?

**Limitations:**

yes

**Strengths And Weaknesses:**

### Strengths

- The central idea of the initialization strategy is appealing and well-motivated. Framing the selection of the adaptation in terms of task-dependent curvature aligns with prior ideas in curvature-aware optimization (e.g., elastic weight consolidation, sharpness-aware minimization).
- The empirical analysis is extensive, capturing different types of NLP tasks (e.g., reasoning, QA), as well as image classification tasks. Moreover, there are plenty of baselines (e.g., standard LoRA and data-free initialization strategies) and similar data-dependent strategies. The LLM suite is also broad, covering various model families.
- The empirical results are strong and appear consistent across tasks, providing large improvements over the standard LoRA initialization, and also marginal improvements over similar methods.
- The method appears time-efficient compared to other data-dependent adaptation strategies.

---

### Weaknesses
- The mathematical exposition is unclear in several places, where some steps are omitted or rely on implicit assumptions. For instance, the transition from eq. (9) to eq. (10) relies on vectorization identities that are not obvious to me. The attached derivation in Appendix A.4 seems imprecise, as the lemma statement does not match the final results obtained in the proof. The proof establishes $\nabla _W L = (\nabla_Y L) X^\top$, which has dimensions $m \times n$. However, the lemma itself states a Kronecker-form expression involving $X \otimes \nabla_Y L$. This object has dimensions $ (nm) \times l^2$, so there appears to be a mismatch without the vectorization steps if I'm not mistaken. Furthermore, the paper introduces the notation of a "symmetric perturbation along direction $Z$" and argues that first-order terms cancel, leaving second-order curvature as the dominant factor. However, the concept of symmetric perturbation is not formally defined, and its precise meaning as well as the derivation remain vague to me.
- The initialization procedure itself is somewhat unclear. In Section 3.2, Eq. (8) is described as the initialization, but the equation appears to just be a decomposition of the base weight matrix. It is unclear how this connects to earlier formulation (whether it is computed as in Eq. (1)) or how the specific singular values used in the initialization are selected. Also, I'm not sure what the index $i$ in $\frac{ih}{32}$ represents.
- The experimental setup in Figure 1 only evaluates Fisher energy as a ranking criterion over singular directions of the pretrained weights. As a result, the figure demonstrates that Fisher energy may rank SVD directions better than singular values, but it does not establish that Fisher-aligned directions themselves are a desirable LoRA adaptation space. Also, the figure is a bit hard to interpret since the plots overlay accuracy values with the underlying metric, while the x-axis represents a sorted experiment index.
- The evaluation lacks several important details. No standard deviations or statistical tests are reported, which makes it hard to assess the robustness of the method. I think that std devs are particularly important since the initialization process can heavily influence the fine-tuning process, which the authors acknowledge and use as the main motivation for the paper.
- The paper does not provide a memory analysis. While the authors acknowledge (in the limitations) that the method introduces additional memory overhead, it would be useful to quantify this overhead in practice.
- The method is inherently limited to situations where at least some downstream data is available, which may restrict its applicability in continual or streaming fine-tuning scenarios. It remains unclear how much data is needed before fine-tuning to achieve a reliable initialization using the proposed approach.
- Some sentences appear to be repeated in close proximity (with slight paraphrasing). There are some minor typos (see comments below).


---

#### Comments
- L043--L044: "FILet compute" => computes
- L080--L081: "yang Liu et al." => Liu et al.?
- L145--L146: "." instead of "," at the end of Eq. (4)
- Eq. (9): $x$ => $X$?
- Sentences in L163--164 and L188--L192 appear repeated with minor paraphrasing.
- Figure 1: perhaps provide a label for the x axis

---

> ### Author Rebuttal · Authors · 2026-03-29
>
> Thank you for your valuable feedback. Below, we provide responses.
>
> 1. Weakness 1
>
> This issue is due to a typo in Appendix A.1 and A.4. The correct formalization in Lemma A.1. is $\nabla_W \mathcal{L} = (\nabla_Y \mathcal{L}) X^\top$. Our derivation is performed at the per-token level: we compute the gradient for each individual token and then average over the sequence and the minibatch. Accordingly, the consistent symbol definition in Appendix A.1 is $X \in \mathbb{R}^{n \times 1}$, $Y \in \mathbb{R}^{m \times 1}$, and $\nabla_Y \mathcal{L} \in \mathbb{R}^{m \times 1}$.  We will correct these typos in the camera-ready version.
>
> 2. Weakness 2, Question 2
>
> This initialization is designed to demonstrate that data sensitivity is the key factor underlying performance differences across singular directions. To this end, we sample 32 groups of singular directions ordered from the largest to the smallest singular values. The variable $i$ denotes the group index. We will refine this in cemera-ready verion.
>
> 3. Weakness 3
>
> We agree that Figure 1 is not meant to demonstrate globally optimal LoRA subspaces. Instead, it shows that, within singular directions, Fisher Energy provides a more informative ranking signal than singular values. The effectiveness of FILet is validated by downstream experiments and ablations, where Fisher-guided initialization consistently improves performance. We will revise the text around Figure 1 to clarify this distinction.
>
> 4. Weakness 4
>
> We reorganize our experimental results and report the standard deviations for FILet and the reproduced baselines in the subscripts.
>
> ||BoolQ|PIQA|SIQA|HellaS.|WinoG.|ARC-e|ARC-c|OBQA|Avg.|
> |-|-|-|-|-|-|-|-|-|-|
> |KaSA(Llama2-7b)|74.3(0.55)|85.2(0.66)|81.9(0.32)|93.4(0.23)|85.3(0.59)|86.3(0.62)|67.4(0.87)|80.2(0.77)|81.7(0.22)|
> |LoRA-Dash(Llama2-7b)|73.9(0.58)|85.0(0.79)|82.0(0.38)|94.2(0.19)|84.3(0.90)|85.7(0.49)|68.6(0.92)|79.4(0.79)|81.7(0.38)|
> |FILet(Llama2-7b)|75.0(0.40)|86.6(0.50)|82.8(0.50)|94.2(0.22)|85.7(0.53)|87.1(0.58)|70.5(0.81)|82.6(0.62)|83.1(0.26)|
> |KaSA(Llama3-8b)|74.3(0.71)|87.8(0.70)|82.2(0.55)|95.2(0.38)|87.5(0.54)|91.6(0.57)|80.3(0.63)|87.0(0.62)|85.7(0.23)|
> |LoRA-Dash(Llama3-8b)|74.7(0.67)|87.4(0.53)|82.0(0.42)|95.8(0.30)|87.2(0.55)|92.0(0.43)|80.0(0.66)|86.0(1.06)|85.6(0.29)|
> |FILet(Llama3-8b)|75.4(0.55)|90.6(0.57)|82.8(0.52)|96.1(0.28)|89.0(0.45)|92.2(0.58)|79.9(0.39)|87.4(0.61)|86.7(0.13)|
>
> 5. Weakness 5
>
> In our experimental setting, the memory requirements (GBs) of LoRA and FILet initialization and training are reported below.
>
> ||BoolQ|PIQA|SIQA|HellaS.|WinoG.|ARC-e|ARC-c|OBQA|
> |-|-|-|-|-|-|-|-|-|
> |LoRA(Llama2-7B)|24.9|52.1|35.2|44.1|26.9|33.8|33.5|29.8|
> |FILet(Llama2-7B)|32.9|65.3|43.0|52.2|37.0|45.5|45.2|38.9|
> |LoRA(Llama3-8B)|26.5|50.3|33.4|52.2|28.3|34.7|35.2|30.3|
> |FILet(Llama3-8B)|36.1|63.0|42.5|62.4|39.3|47.3|47.8|41.5|
>
> FILet uses more memory during initialization to cache intermediate values but consistently delivers much better performance.
>
> 6. Weakness 6
>
> In our NLG experiments, MBPP and MT-Bench lack training sets, so we use related data for initialization (CodeFeedback for MBPP, Alpaca for MT-Bench). Results show that exact dataset matching is unnecessary as long as the domain is similar.
>
> From Section 4.5.1, smaller minibatches (80) yield better average performance, while a size around 320 offers a better trade-off between performance and initialization time.
>
> 7. Weakness 7, Question 3
>
> We apologize for the limited space in the rebuttal. In the camera-ready version, we will fix all typos and revise Eq. (8) to match the form of Eq. (1).
>
> 8. Question 1
>
> $Z$ in our problem definition is a rank-1 direction, here is the formal definition:
> $$
> Z := \textbraceleft u v^\top | u \in \mathbb{R}^m, v \in \mathbb{R}^n, \|u\|_2 = 1,\;\|v\|_2 = 1 \textbraceright.
> $$
>
> The first-order term in Eq. (3) does not generally cancel, but we intentionally focus on the second-order term. From an initialization perspective, LoRA splits pretrained weights into a fixed part and a trainable part, aiming to preserve important knowledge while updating less useful directions. In Eq. (2), we avoid directions that significantly increase loss, as they likely contain important information.
>
> The first-order term reflects the gradient, which is naturally followed during fine-tuning and thus need not be handled at initialization. Instead, the second-order term captures sensitivity. By selecting low-sensitivity directions, we identify a subspace with less task-relevant information, reducing the risk of increasing loss. Hence, the first-order term is not canceled but deliberately ignored to align with the goal of LoRA initialization.
>
> 9. Question 4
>
> We agree that this is a promising direction for future work, aiming to extend Fisher-aligned methods to a new setting. In principle, it is feasible, but it would require periodically re-estimating Fisher statistics and updating the LoRA basis during training, adding computational and implementation overhead.

---

> > ### Author Rebuttal · Reviewer_tkuh · 2026-04-05
> >
> > Thank you for the detailed rebuttal. Most of my concerns have been addressed satisfactorily. The reported std devs confirm that the improvements are consistent, and the memory overhead is now quantified. My remaining concern is with Q1: the rebuttal clarifies that the first-order term is deliberately ignored rather than formally canceled, which differs from the impression given in the paper.
> >
> > Overall, given that the main empirical and presentation concerns have been addressed, I am raising my score to 4.

---

> > > ### Author Response · Authors · 2026-04-05
> > >
> > > We thank the reviewer for pointing this out and apologize for the lack of clarity due to space limitations. We agree that this is a valid concern and the statements in Section 3.1 may be misleading. Below, we provide a more detailed response to Q1:
> > >
> > > Our primary goal in Sections 3.1 and 3.2 is to justify:
> > >
> > > (1) Why do we aim to select directions with low (rather than high) Fisher energy, and
> > >
> > > (2) why the Fisher energy is defined as $\mathcal{E}(Z) = \mathrm{vec}(Z)^\top S_W\,\mathrm{vec}(Z)$
> > >
> > > Our intention is not to strictly minimize Eq.~(2), but to use the Taylor expansion as a motivation to separate two roles in fine-tuning: The first-order term captures the descent direction, while the second-order term reflects the sensitivity (curvature) of the loss landscape.
> > >
> > > In the fine-tuning stage, parameter updates naturally follow the negative gradient direction. Therefore, in the initialization stage, our focus is to identify a subspace with low sensitivity, such that updates within this subspace are less likely to incur large loss increases. This motivates selecting directions with low curvature (low Fisher energy), which can lead to more stable and effective optimization trajectories, as supported by the experiments in Section 3.2.
> > >
> > > The perturbation along direction $Z$ is primarily used in the initialization stage. At this stage, our objective is to exploit structural information of the loss landscape that cannot be captured by standard gradient-based optimization. This motivates us to focus on the second-order term.
> > >
> > > **Revised interpretation.**
> > >
> > > To clarify this point, we will revise the derivation to introduce the symmetric perturbation explicitly.
> > >
> > > Applying a second-order Taylor expansion of the loss around $W_0$ yields Eq. (3). To mitigate the influence of the gradient direction and better measure sensitivity along direction $Z$, we consider a symmetric perturbation, leading to:
> > > $$
> > > \Delta \mathcal{L}_{\text{sym}}(Z) := \frac{1}{2}[ \mathcal{L}(W_0 + \gamma Z) + \mathcal{L}(W_0 - \gamma Z) ] - \mathcal{L}(W_0).
> > > $$
> > >
> > > Under this symmetric construction, the first-order terms cancel out by design, yielding
> > > $$
> > > \Delta \mathcal{L}_{\text{sym}}(Z) = \frac{\gamma^2}{2} < Z, (\nabla_{W_0}^2 \mathcal{L}) Z > + o(\gamma^2).
> > > $$
> > >
> > > This formulation isolates the second-order curvature of the loss landscape, yielding a direction-wise sensitivity measure independent of the downstream gradient. We hope this revision addresses your concern.
> > >
> > > We will revise the manuscript to
> > >
> > > (1) rephrase the statement about first-order cancellation,
> > >
> > > (2) clarify that focusing on the second-order term is a design choice motivated by subspace-constrained optimization rather than a direct consequence of Eq.~(2),
> > >
> > > (3) improve the overall presentation to clearly distinguish between the role of initialization and the role of optimization dynamics.

---

### Decision · Program_Chairs · 2026-04-30

**Decision:**

Accept (regular)

**Comment:**

This paper proposes a Fisher-informed initialization strategy for LoRA fine-tuning. Reviewers generally agreed that the paper addresses an important problem and offers a practically useful and sufficiently novel contribution. A major strength of the submission is its broad empirical evaluation across multiple tasks, model families, and baselines, which provides convincing evidence of the method’s effectiveness. During rebuttal, the authors addressed several reviewer concerns by adding variance statistics and clarifying the efficiency and memory trade-offs. The main remaining issue is the theoretical presentation, which would benefit from a more careful and modest framing.
Overall, I recommend weak acceptance. If accepted, I would encourage the authors to revise the theoretical framing to avoid overstating what is formally justified.